# The natverse, a versatile toolbox for combining and analysing neuroanatomical data

Alexander Shakeel Bates[1†], James D Manton[1†], Sridhar R Jagannathan[2], Marta Costa[1,2], Philipp Schlegel[1,2], Torsten Rohlfing[3], Gregory SXE Jefferis[1,2]*

[1]Neurobiology Division, MRC Laboratory of Molecular Biology, Cambridge, United Kingdom; [2]Drosophila Connectomics Group, Department of Zoology, University of Cambridge, Cambridge, United Kingdom; [3]SRI International, Neuroscience Program, Center for Health Sciences, Menlo Park, United States

**Abstract** To analyse neuron data at scale, neuroscientists expend substantial effort reading documentation, installing dependencies and moving between analysis and visualisation environments. To facilitate this, we have developed a suite of interoperable open-source R packages called the `natverse`. The `natverse` allows users to read local and remote data, perform popular analyses including visualisation and clustering and graph-theoretic analysis of neuronal branching. Unlike most tools, the `natverse` enables comparison across many neurons of morphology and connectivity after imaging or co-registration within a common template space. The `natverse` also enables transformations between different template spaces and imaging modalities. We demonstrate tools that integrate the vast majority of *Drosophila* neuroanatomical light microscopy and electron microscopy connectomic datasets. The `natverse` is an easy-to-use environment for neuroscientists to solve complex, large-scale analysis challenges as well as an open platform to create new code and packages to share with the community.

**\*For correspondence:**
jefferis@mrc-lmb.cam.ac.uk

[†]These authors contributed equally to this work

**Competing interests:** The authors declare that no competing interests exist.

## Introduction

Neuroanatomy has become a large-scale, digital and quantitative discipline. Improvements in sample preparation and imaging increasingly enable the collection of large 3D image volumes containing complete neuronal morphologies in the context of whole brains or brain regions. Neuroscientists, therefore, need to tackle large amounts of morphological data, often writing custom code to enable repeated analysis using their specific requirements. They also need to analyse neuronal morphology and connectivity in the context of whole nervous systems or sub-regions. However, it is desirable not to rewrite basic functionalities such as reading various types of data file, representing neurons in different data structures, implementing spatial transforms between samples, integrating popular datasets or performing popular analyses from scratch. Scaling up or developing custom analysis strategies is simpler and more feasible for researchers if they can reuse existing infrastructure. This has been amply demonstrated by flexible but open source platforms such as ImageJ/Fiji for image analysis (*Schindelin et al., 2012*) or Bioconductor for bioinformatics (*Huber et al., 2015*). One important consequence of these free and open-source tools is that they aid collaboration and reproducibility, and reduce the overhead when switching between different types of analysis. Together, these considerations have motivated us to create the NeuroAnatomy Toolbox (`nat`) and its extensions, which we detail in this paper.

A number of software tools are already available to analyse neuronal data (*Billeci et al., 2013*; *Brown et al., 2005*; *Cuntz et al., 2010*; *Feng et al., 2015*; *Gensel et al., 2010*; *Glaser and Glaser, 1990*; *Ho et al., 2011*; *Katz and Plaza, 2019*; *Kim et al., 2015*; *Meijering et al., 2004*;

*Myatt et al., 2012*; *Narro et al., 2007*; *Peng et al., 2014*; *Pool et al., 2008*; *Saalfeld et al., 2009*; *Schmitz et al., 2011*; *Wearne et al., 2005*). However, most focus on image processing and the morphological analysis options available are fairly basic, such as examining arbour lengths or performing Sholl analyses (*Sholl, 1953*). Of these, the trees toolbox (*Cuntz et al., 2010*), has particularly strong support for morphological analysis of neurons but focuses on individual neurons in isolation rather than neurons within the volume of the brain as a whole.

Recent technological advances have made acquiring large amounts of neuronal morphology data in their whole-brain contexts feasible across phyla (*Chiang et al., 2011*; *Cook et al., 2019*; *Economo et al., 2016*; *Jenett et al., 2012*; *Kunst et al., 2019*; *Li et al., 2019*; *Oh et al., 2014*; *Ohyama et al., 2015*; *Ryan et al., 2016*; *Winnubst et al., 2019*; *Zheng et al., 2018*). Image data are typically registered against a template space, allowing one to compare data from many brains directly and quantitatively. This significantly aids the classification of neuronal cell types because it allows type classification relative to the arbours of other neuronal types (*Sümbül et al., 2014*) and anatomical subvolumes. However, while this enables the comparison of data within a given study, template spaces are often different across studies or laboratories, hindering data integration.

This paper describes the Neuroanatomy Toolbox (`nat`), a general purpose open source R-based package for quantitative neuroanatomy, and a suite of extension R packages that together we call the `natverse`. A distinctive feature of the `natverse`, as compared with other available tools, is to analyse neurons within and across template spaces and to simplify access to a wide range of data sources. Neurons can be read from local files or from online repositories (*Ascoli et al., 2007*; *Chiang et al., 2011*; *Economo et al., 2016*; *Jenett et al., 2012*; *Kunst et al., 2019*; *Winnubst et al., 2019*) and web-based reconstruction environments (*Katz and Plaza, 2019*; *Saalfeld et al., 2009*; *Schneider-Mizell et al., 2016*). The `natverse` can be installed in two lines of code as described on the project website (https://natverse.org). Every function is documented with a large number of examples based on bundled or publicly available data. Example pipeline code, and code to generate the figures in this manuscript is available through https://github.com/natverse/nat.examples. We provide online community support through our nat-user mailing list: https://groups.google.com/forum/#!forum/nat-user.

The `natverse` has recently been employed for large-scale analysis of zebrafish data (*Kunst et al., 2019*), and we provide examples across a range of invertebrate and vertebrate species. We then give more specific examples focussing on cell type identification across *Drosophila* datasets. Using the `natverse`, we have created bridging registrations that transform data from one template to another along with mirroring registrations (e.g. left-to-right hemisphere) and made these easily deployable. This unifies all publicly available *Drosophila* neuroanatomical datasets, including those image data for genetic resources and whole brain connectomics.

We now give an overview of the `natverse` and showcase a number of common applications. These applications include quantifying the anatomical features of neurons, clustering neurons by morphology, analysing neuroanatomical data relative to subvolumes, in silico intersections of genetic driver lines, matching light-level and EM-level neuronal reconstructions and registering and bridging neuroanatomical data to and between template spaces.

## Results

### Software packages for neuroanatomy

We have opted to develop our software in R, a leading platform for bioinformatics and general data analysis. R is free and open source, and is supported by high-quality integrated development environments (e.g. Rstudio). It features a well-defined system for creating and distributing extension packages that bundle code and documentation. These can easily be installed from high-quality curated repositories (CRAN, Bioconductor) as well as via GitHub. R supports a range of reproducible research strategies including reports and notebooks and integrates with the leading cross-platform tools in this area (jupyter, binder).

The core package of the `natverse` is the Neuroanatomy Toolbox, `nat`. It supports 3D visualisation and analysis of neuroanatomical data (*Figure 1a*), especially tracings of single neurons (*Figure 1b*). `nat` allows a user to read neuronal data from a variety of popular data formats produced by neuron reconstruction tools (*Figure 1a*). Typical image analysis pipelines include imaging

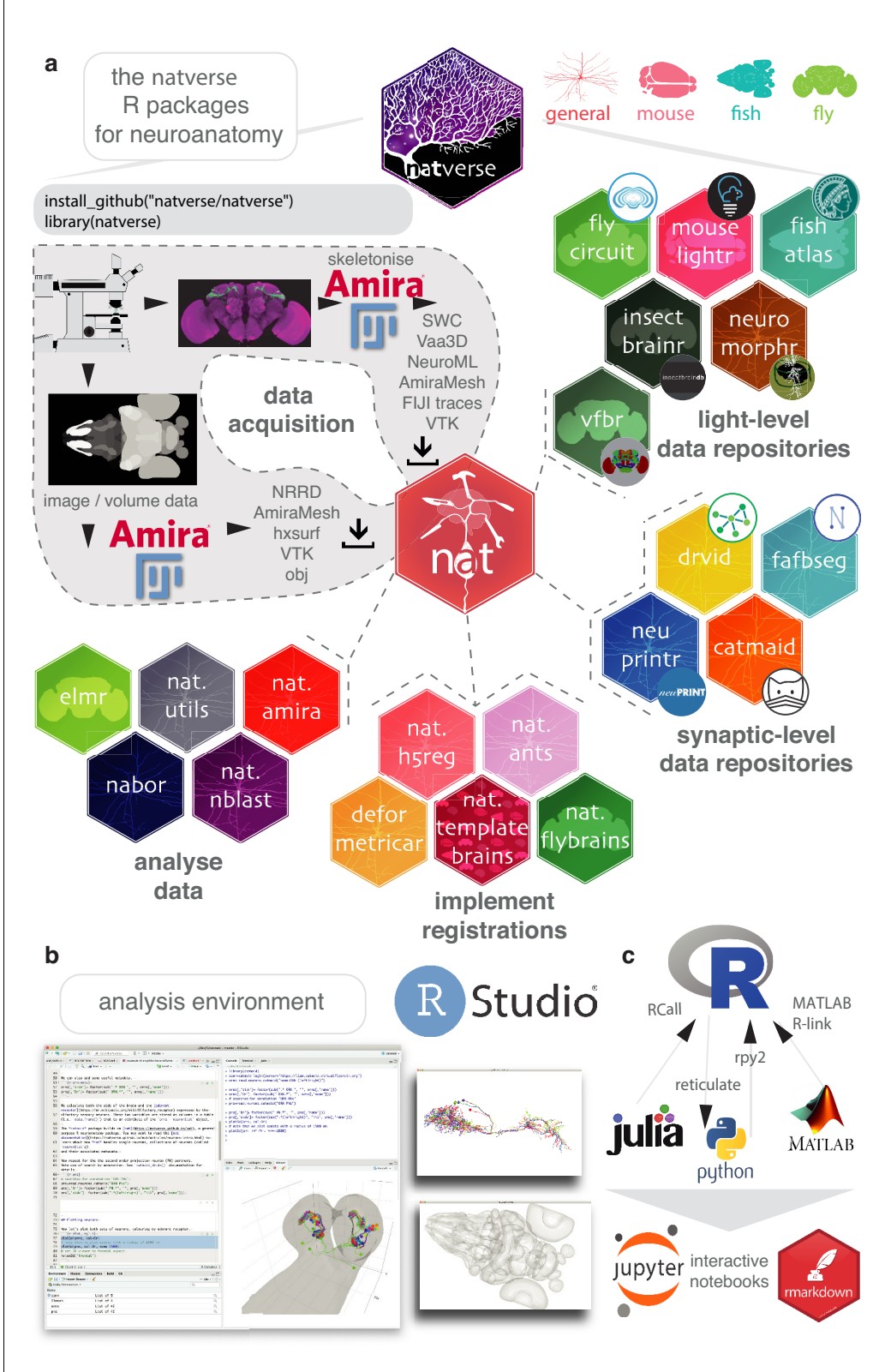

**Figure 1.** The natverse. (**a**) R packages that constitute the `natverse`. Packages are coloured by whether they are general purpose, or cater specifically for *Mus musculus*, *Danio rerio* or *Drosophila melanogaster* datasets. Coarse division into packages for fetching remote data, implementing registrations and analysing data are shown. Data, as outputted by most reconstruction pipelines, can also be read by `nat`. (**b**) The `natverse` is designed to work best in the RStudio environment (**RStudio Team, 2015**), by far the most popular environment in which to execute and write R code. 3D visualisation is

*Figure 1 continued on next page*

*Figure 1 continued*

based on the R package rgl (*Murdoch, 2001*), which is based on OpenGL and uses the XQuartz windowing system. It runs on Mac, Linux and Windows platforms. (c) R functions can be called by other popular scientific programming languages, some example packages/libraries are shown. In particular, there is support for bidirectional interaction with Python, with interactive use supported by Jupyter or R Markdown notebooks. One of us (P. Schlegel) has developed Python code inspired by the `natverse` for analysing neuron morphology, NAVIS (https://github.com/schlegelp/navis), and talking to the CATMAID API, PyMaid (https://github.com/schlegelp/pyMaid). These python libraries transform `natverse`-specific R objects into Python data types and can call `natverse` R functions like `nblast`.

The online version of this article includes the following figure supplement(s) for figure 1:

**Figure supplement 1.** A basic analysis pipeline.

---

neurons with confocal microscopy, reconstructing them using Fiji Simple Neurite Tracer (*Longair et al., 2011*) then saving them as SWC files (*Cannon et al., 1998*); `nat` can read a collection of such files with a single command. In addition, a user can, for example, mark the boutons on each neuron using Fiji's point tool and export that as a CSV, load this into nat and then analyse the placement of these synaptic boutons with respect to the originally traced neuron (*Figure 1—figure supplement 1*).

We have extended `nat` by building the `natverse` as an ecosystem of interrelated R packages, each with a discrete purpose (*Figure 1a*). The `natverse` is developed using modern software best practices including revision control, code review, unit testing, continuous integration, and comprehensive code coverage. Developing sets of functions in separate packages helps compartmentalise development, ease troubleshooting and divides the `natverse` into documented units that users can search to find the more specific code examples or functions that they need. To the casual user, these divisions may initially be of little consequence. We therefore provide a single wrapper package, `natverse`; installing this results in the installation of all packages and their dependencies, immediately giving the user all the capabilities described in this paper (*Figure 1a*). Natverse packages have already been used in recent publications from our lab (*Cachero et al., 2010*; *Costa et al., 2016*; *Dolan et al., 2019*; *Dolan et al., 2018a*; *Dolan et al., 2018b*; *Frechter et al., 2019*; *Grosjean et al., 2011*; *Huoviala et al., 2018*; *Jefferis et al., 2007*) and others (*Clemens et al., 2018*; *Clemens et al., 2015*; *Eichler et al., 2017*; *Felsenberg et al., 2018*; *Jeanne et al., 2018*; *Kunst et al., 2019*; *Saumweber et al., 2018*; *Zheng et al., 2018*), with the nat.nblast packaged described in *Costa et al., 2016*. Confirmed stable versions of `nat`, `nat.templatebrains`, `nat.nblast`, `nat.utils` and `nabor` can be downloaded from the centralised R package repository, CRAN, with developmental versions available from our GitHub page (https://github.com/natverse/).

In brief, `natverse` packages can be thought of as belonging to four main groups (*Figure 1a*). The first two support obtaining data, either by a) interacting with repositories and software primarily used for neuron reconstructions from electron micrograph (EM) data, including CATMAID, NeuPrint and DVID (*Clements et al., 2020*; *Katz and Plaza, 2019*; *Saalfeld et al., 2009*; *Schneider-Mizell et al., 2016*) or b) interacting with repositories for light-level data, including MouseLight, FlyCircuit, Virtual Fly Brain, NeuroMorpho, the InsectBrainDB and the FishAtlas projects.

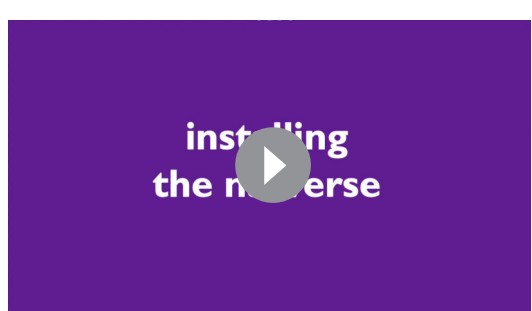

**Video 1.** Short tutorial videos. Short tutorial on how to use basic `natverse` functionality in RStudio, for example, loading and installing the natverse, plotting neurons and volumes, bridging between template brains, using NBLAST and comparing EM and LM data.
https://elifesciences.org/articles/53350#video1

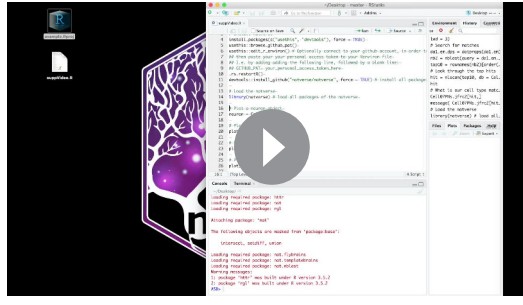

**Video 2.** Installing the natverse.
https://elifesciences.org/articles/53350#video2

Additional R packages help with c) manipulating and deploying registrations to move data between brainspaces, and d) data analysis and visualisation (see Materials and methods for additional details). In order to see how one can use the `natverse` in RStudio to visualise and analyse neurons, please see *Videos 1–5*.

## Manipulating neuroanatomical data

### Neuron skeleton data

Raw 3D images enable true to life visualisation but simplified representations are usually required for data analysis. For example, neurons can be traced to generate a compact 3D skeleton consisting of 3D vertices joined by edges. A more accurate representation would be a detailed mesh describing a 3D neuron, but it is often easier and quicker to work with skeleton representations.

The `natverse` provides functions for morphological and graph-theoretic analyses of neurons, collections of neurons, neurons as vector clouds and neurons as tree graphs (*Figure 2a*). The `natverse` mainly operates with skeleton data, but the geometry of neuron mesh data can be analysed using the more general R packages `Rvcg` and `Morpho` (*Schlager, 2017*). The `natverse` represents skeletonised neurons as `neuron` objects, with the neuron's geometry in the standard SWC format where each node in the skeleton has its own integer identifier. There are additional data fields (*Figure 2—figure supplement 2*), the treenode IDs for branch points, the location of its synapses in 3D space and their polarity, including the source file, leaf nodes and series of IDs that belong to continuous non-branching segments of the neuron (*Figure 2—figure supplement 2*).

Neurons have tree like structures that are critical to their function (*Cuntz et al., 2010*). `ngraph` data objects represent a neuron as a tree graph originating at a root (usually the soma) with directed edges linking each of the neuron's tree nodes (*Figure 2a*). This representation provides a bridge to the rich and efficient graph theory analysis provided by the `igraph` package (*Csardi and Nepusz, 2006*).

Objects of class `neuron` are lists of data objects, like `data.frames`, describing properties such as the 3D position and interconnectivity of points in a neuron. Objects of class `neuronlist` are lists of `neuron` objects, representing one or more neurons, with some attached metadata. This attached metadata can give information like a neuron's name, some unique identifier, its cell type, etc (*Figure 2—figure supplement 2*). An `ngraph`, `neuron` or `neuronlist` can be passed to many functions in the `natverse`, and also to other functions familiar to R users for which we have written specific methods. For example, users can call `subset` on a `neuronlist` to select only those neurons with a certain entry in their corresponding metadata, for example all amacrine cells. Methods passed to `plot3d` enable a `neuronlist` to be coloured by its metadata entries when it is plotted (*Figure 2b*), in this case connectomic data from the inner plexiform layer of the mouse retina is shown (*Helmstaedter et al., 2013*). Many functions are built to work with `neuron` objects but will also have a method that allows them to be applied to every `neuron` in a given `neuronlist` via the `nat` function `nlapply`. R users will be familiar with this logic from using the base R function `lapply`.

### Basic analysis

A useful function with methods for `neuron` objects and `neuronlist` objects is `summary`. This gives the user counts for tree nodes, branch points, leaf nodes and the total combined cable length of a neuron (*Figure 2—figure supplement 1a*). We can further use the `natverse` to identify points on a neuron that have particular properties based on the neuron's skeleton structure (*Figure 2c–e*) or because we have some other data that identifies the position of some biological feature (*Figure 2f–g*), or both (*Figure 2h*). Branching can be assessed by branching density, for example a Sholl analysis (`sholl_analysis`) (*Figure 2—figure*

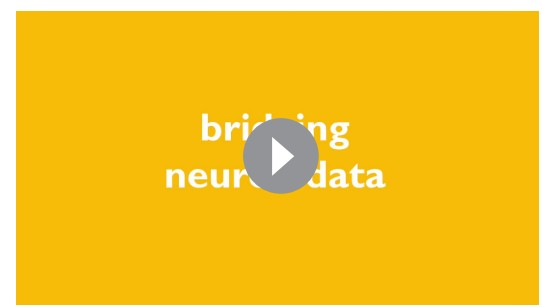

**Video 3.** Bridging neuron data.
https://elifesciences.org/articles/53350#video3

*supplement 1b*), or decomposed by branching complexity, for example Strahler order (*Figure 2—figure supplement 1c*). Geodesic distances, that is within-skeleton distances, can be calculated between any tree node in the graph (*Figure 2—figure supplement 1c*) with the help of functions from the R package `igraph` (*Csardi and Nepusz, 2006*), and Euclidean distances can be calculated using our R package `nabor`.

Some reconstruction environments allow tree nodes to be tagged with extra information, for example CATMAID. This can include neurite diameter, microtubules (*Figure 2e*) and pre- and postsynapses (*Figure 2f*). This information is fetched when the `catmaid` package reads a neuron. It can be used by a graph theoretic algorithm (*Schneider-Mizell et al., 2016*; *Figure 2g*, inset) to divide a neuron into its dendrites, axon and intervening cable (*Figure 2h*). We put this information together in the example in *Figure 2—figure supplement 1c*, which shows the geodesic distribution of pre- and postsynapses along three neurons arbors, split by axon and dendrite, then further by Strahler order, then further by presence or absence of microtubule. Here, for our three exemplar neurons, presynapses only exist on microtubular backbones, and are laid in high number except at the highest Strahler orders while postsynapses are mainly on twigs, and at Strahler order 1–2. We can also identify connected neurons using `catmaid` functions, and see that the dendrites of these cells only receive particular inputs.

## Neuroanatomical volumes

The `natverse` also helps users to analyse neuronal skeletons with respect to volume objects that might represent neuroanatomical structures on a scale from whole neural tissues to neuropil subvolumes. 3D objects from diverse sources can be visualised and analysed with `nat`, and we can calculate their volumes (*Figure 3a*). By using the `nat` function `make_model`, a user can interactively create their own 3D objects from, for example, 3D points from a neuron's cable or its synapses (*Figure 3b*); points can easily be retrieved by giving the function a labelled `data.frame`, `matrix`, `neuron`, `neuronlist`, `hxsurf` or `mesh3d` object (*Figure 2—figure supplement 2*). The resulting volume could be, for example, the envelope around a dendrite, which may correlate with other features of a neuron (*Figure 3b*). Using the `nat` function `prune_in_volume`, a skeleton can be cut to include or exclude the cable within a given volume, while the function `pointsinside` can tell a user which synapses lie within a standard neuropil segmentation (*Figure 3c*).

## Advanced analysis

Because the `natverse` is a flexible platform that allows users to easily write their own R code to support intricate procedures, very specific analyses can be performed. For example, we might be interested in using skeletons to define anatomical subvolumes and analysing the projections between such subvolumes. For *Figure 3—figure supplement 1*, we developed custom code on top of `natverse` functionality to examine light-level *D. melanogaster* olfactory projections to, and target neurons with dendrites in a subregion of the brain called the lateral horn (*Chiang et al., 2011*; *Frechter et al., 2019*; *Grosjean et al., 2011*). We voxelised the lateral horn as well as its target regions into overlapping kernel density estimates based on agglomerating similarly shaped subbranches for projection neuron axons. This analysis reveals substructure in a neuropil, and the 3D locations that are likely to receive input from these new subregions (*Figure 3—figure supplement 1d*). The `natverse` contains other functions to infer connectivity from light-level data, including `potential_synapses`, an implementation of a synapse prediction algorithm that makes use of spatial proximity and approach angle (*Stepanyants and Chklovskii, 2005*), and `overlap`, a simpler algorithm that measures the putative overlap in Euclidean space between neuron pairs (*Frechter et al., 2019*).

## Cell typing neurons

Neuronal cell type is a useful classification in neuroscience (*Bates et al., 2019*). Neuronal cell typing can be done by expert examination (*Helmstaedter et al., 2013*), purely by morphological clustering (*Jeanne and Wilson, 2015*), or a combination of both (*Frechter et al., 2019*). Many neurogeometric algorithms for assessing similarity exist. Some are invariant to the 3D embedding space (*Li et al., 2017*; *Sholl, 1953*; *Wan et al., 2015*), but those that are dependent on neurons' relative positioning in a template space have typically met with greater success (*Li et al., 2017*; *Zhao and Plaza, 2014*).

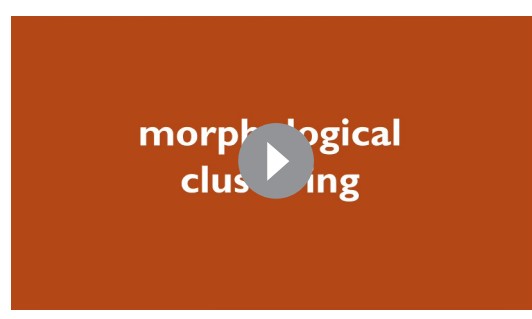

**Video 4.** Morphological clustering.
https://elifesciences.org/articles/53350#video4

NBLAST (*Costa et al., 2016*) is a recent morphological similarity algorithm (*Frechter et al., 2019*; *Jeanne et al., 2018*; *Kohl et al., 2013*; *Kunst et al., 2019*; *Masse et al., 2012*; *Strutz et al., 2014*; *Zheng et al., 2018*). NBLAST is included in the `natverse` in our `nat.nblast` package (*Costa et al., 2016*).

In many parts of mammalian nervous systems, morphologically similar neurons are repeated in space, and so aligning neurons to one another, without a specified template space, is sufficient for quantitative comparison (*Figure 4a*). NBLAST scores can be hierarchically clustered in R, plotted as a dendrogram, and used to visualize morphological groups at a defined group number or cut height (*Figure 4a*). Often, this forms a good starting point for cell typing, but might not be in exact agreement with manually defined cell types (*Figure 4b*). This can be due to neuron reconstructions being differently severed by the field of view or size of the tissue sample collected (*Helmstaedter et al., 2013*), or due to registration offsets between registered neuronal skeletons (*Chiang et al., 2011*; *Kunst et al., 2019*). The `natverse` includes interactive functions, such as `nlscan`, that allow users to visually scan neurons and identify mis-assignments (*Figure 4c*), or `find.neuron` and `find.soma`, that allow users to select a neuron of interest from a 3D window (*Figure 4c*).

In smaller brains, like insect central brains or larval fish central brains, the overlap of both axons and dendrites in 3D space is an excellent starting point for defining a neuronal type, since neurite apposition is suggestive of synaptic connectivity (*Rees et al., 2017*) and neurites are highly stereotyped (*Jenett et al., 2012*; *Pascual et al., 2004*). If they have been registered to whole brain templates (*Chiang et al., 2011*; *Costa et al., 2016*; *Kunst et al., 2019*), it is desirable to choose a canonical brain hemisphere and standardise such that all neurons are mapped onto this side to approximately double the neurons available for clustering and assign the same cell types on both hemispheres (*Figure 4—figure supplement 1*, *Figure 4—figure supplement 2*).

## Comparing disparate datasets
### Template brains in *D. melanogaster*

It is highly desirable to compare neurons between datasets within a singular template space. Considering just the case of *D. melanogaster*, separate template brains 'contain' many large and useful but disparate datasets (*Table 1*):~23,000 single light-level neuronal morphologies, hundreds of neuronal tracings from dye fills, a collection of ~11,000 genetic driver lines, ~100 neuroblast clones, and connectomic data, including a brainwide draft connectome on the horizon (*Scheffer and Meinertzhagen, 2019*; *Zheng et al., 2018*). Because of the wealth of data available for *D. melanogaster*, we focus on its brain for our registration examples.

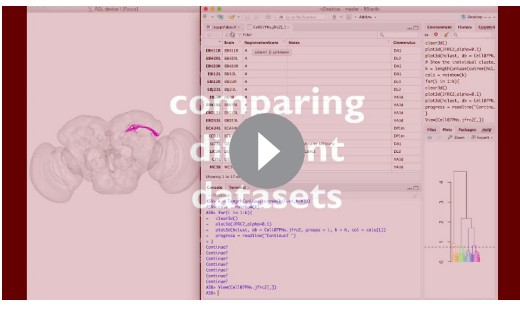

**Video 5.** Comparing different datasets.
https://elifesciences.org/articles/53350#video5

Two approaches have been taken in specifying template spaces: a) choosing a single brain avoids any potential artifacts generated by the averaging procedure, but b) an average brain can reduce the impact of biological variation across individuals and deformations introduced during sample preparation, thus increasing the likelihood of successful and accurate registration (*Bogovic et al., 2018*). Quantitative neuroanatomical work requires images to be spatially calibrated (i.e. with an accurate voxel size), but such calibrations are not present in all template brains.

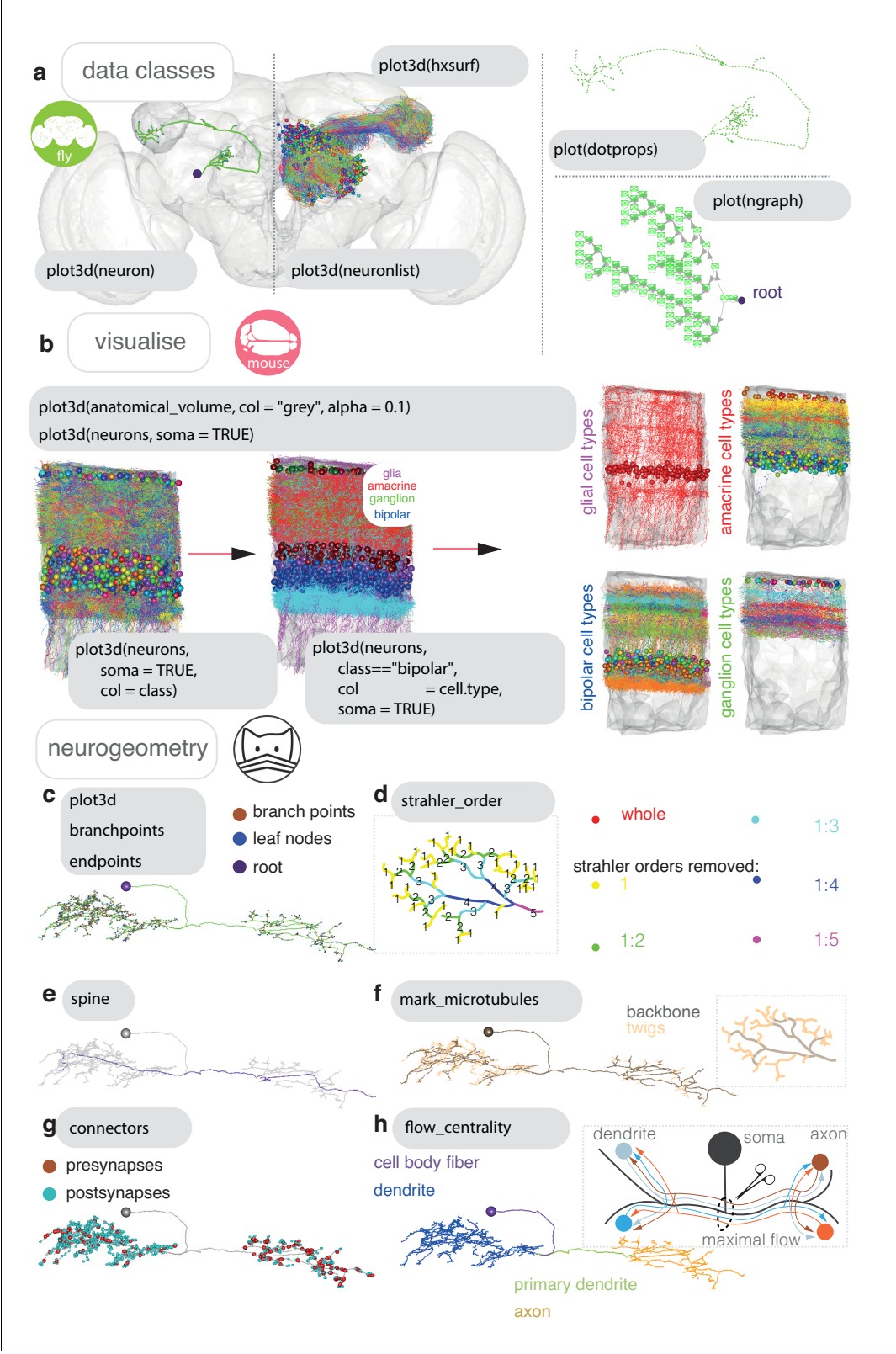

**Figure 2.** Neurons in nat. (a) Data classes defined by `nat`. A *D. melanogaster* DA1 olfactory projection neuron (***Costa et al., 2016***) is shown as part of four different data types, useful for different sorts of analyses: as a `neuron` object (left), as part of a larger `neuronlist` of multiple olfactory projection neurons (middle), as a vector-cloud `dotprops` object (right, upper) and a `ngraph` object (right, lower). In grey, the FCWB template brain, a `hxsurf` object from the package `nat.flybrains`, is shown. Generic code for visualizing these data types shown. (b) Visualisation, with generic sample code,

*Figure 2 continued on next page*

*Figure 2 continued*

of connectomic data from a dense reconstruction inner plexiform layer of the mouse retina is shown, coloured by the cell class and then cell type annotations given by their reconstructors (*Helmstaedter et al., 2013*). Because this dataset contains many neuron fragments that have been severely transected, we only consider skeletons of a total cable length greater than 500 µm using functions `summary` and `subset`. Somata are shown as spheres. (c) A synaptic-resolution neuron reconstruction for a *D. melanogaster* lateral horn neuron (*Dolan et al., 2018a*) has been read from a live CATMAID project hosted by Virtual Fly Brain (https://fafb.catmaid.virtualflybrain.org/) using `catmaid`, and plotted as a `neuron` object. It is rooted at the soma, consistent with the convention. (d) Boxed, Strahler order is a measure of branching complexity for which high Strahler order branches are more central to a neuron's tree structure, and the lower order ones more peripheral, such that branches with leaf nodes are all Strahler order 1. Main, the same neuron which has had its lower Strahler order branches (see inset) progressively pruned away. (e) We can extract the longest path through a neuron, its 'spine', purple, a method that could help define the tracts that a neuron might traverse. (f) Boxed, in insect neurons, the main structure of the neuron is supported by a microtubular backbone. As it branches its more tortuous, smaller caliber neurites loose the microtubule, and make more postsynapses (*Schneider-Mizell et al., 2016*). Main, in CATMAID users can tagged the tree nodes that mark the position where neurite loses its microtubular backbone, so an R user can use `prune` family functions to remove, add or differentially colour microtubular backbone versus twigs. (g) Both presynapses and postsynapses can been manually annotated in CATMAID, and be visualised in R. Because neurons from CATMAID have synaptic data, they are treated as a slightly different class by the natverse, called `catmaidneuron`. A `neuronlist` can also comprise many `catmaidneuron` objects. (h) Right, using synaptic information, it is possible to use a graph theoretic approach which divides the neuron at the point of maximum 'flow' - the region in the neuron at which there are the most parallel paths - having 'drawn' a path between each input synapse and each output synapse that pass through every node on the skeleton (*Schneider-Mizell et al., 2016*). This helps divide a neuron into its dendrites, axon, intervening cable (maximum flow, the primary dendrite) and its cell body fiber (no flow). In insects, the cell body lies outside the neuropil and is connected to its arbour by a single fiber. Main, axon-dendrite split shown for exemplary neuron using `seesplit3d`.

The online version of this article includes the following figure supplement(s) for figure 2:

**Figure supplement 1.** Neurogeometry and skeleton annotations with nat.
**Figure supplement 2.** Neuron data structure.

---

*Table 2* lists the template brains for *D. melanogaster* considered in this work and details the resources available for each; some are shown in Figure 6. Initially, only raw unregistered data were publicly available for FlyCircuit (*Chiang et al., 2011*). Subsequently data registered to one of two template brains (one for each sex). The FlyLight project provides only raw image data (*Jenett et al., 2012*).

Template brains and registered data are publicly available for the Vienna Tiles GAL4 libraries (*Tirian and Dickson, 2017*) but are not distributed in bulk form. We created an intersex reference brain for the FlyCircuit dataset and added spatial calibrations and re-registered data to our new template brains as necessary (see Materials and methods) before constructing bridging registrations. We have deposited all template brain images, in NRRD format (http://teem.sourceforge.net/nrrd/) at http://zenodo.org to ensure long-term availability. Two spatial transforms are most useful when considering template brains - a) mirroring data left-right, so that neurons reconstructed or registered to either hemisphere may be compared, and b) bridging between these templates, to cross-compare data.

## Mirroring data in *D. melanogaster*

Whilst the *Drosophila* brain is highly symmetric it is not perfectly so and the physical handling of brains during sample preparation introduces further non-biological asymmetries. A simple 180° flip about the medio-lateral axis is therefore insufficient (*Figure 5—figure supplement 1a*). To counter this, we have constructed non-rigid warping registrations for a number of template spaces that introduce the small displacements required to fix the mapping from one hemisphere to the other (*Figure 5—figure supplement 1*, see Materials and methods).

Our mirroring registrations can be deployed using the function `mirror_brain`. Our mirroring registrations can be used to counter non-biological asymmetries, allowing the investigation of relevant similarities and differences in morphology between the two sides of the brain (*Figure 5—figure supplement 1a*). Our mirroring procedure (see Materials and methods) does not introduce any systematic errors into neuron morphology.

NBLAST was used to calculate morphologically determined similarity scores between DL2d projection neurons taken from the same side of the brain and compare them with those calculated between DL2d projection neurons taken from alternate sides of the brain (*Figure 5b*). We do not find the distributions of scores (*Figure 5c*) to be significantly different (D = 0.025, p=0.094, two-sample Kolmogorov-Smirnov test). Extending this, we have used these scores to classify neurons based

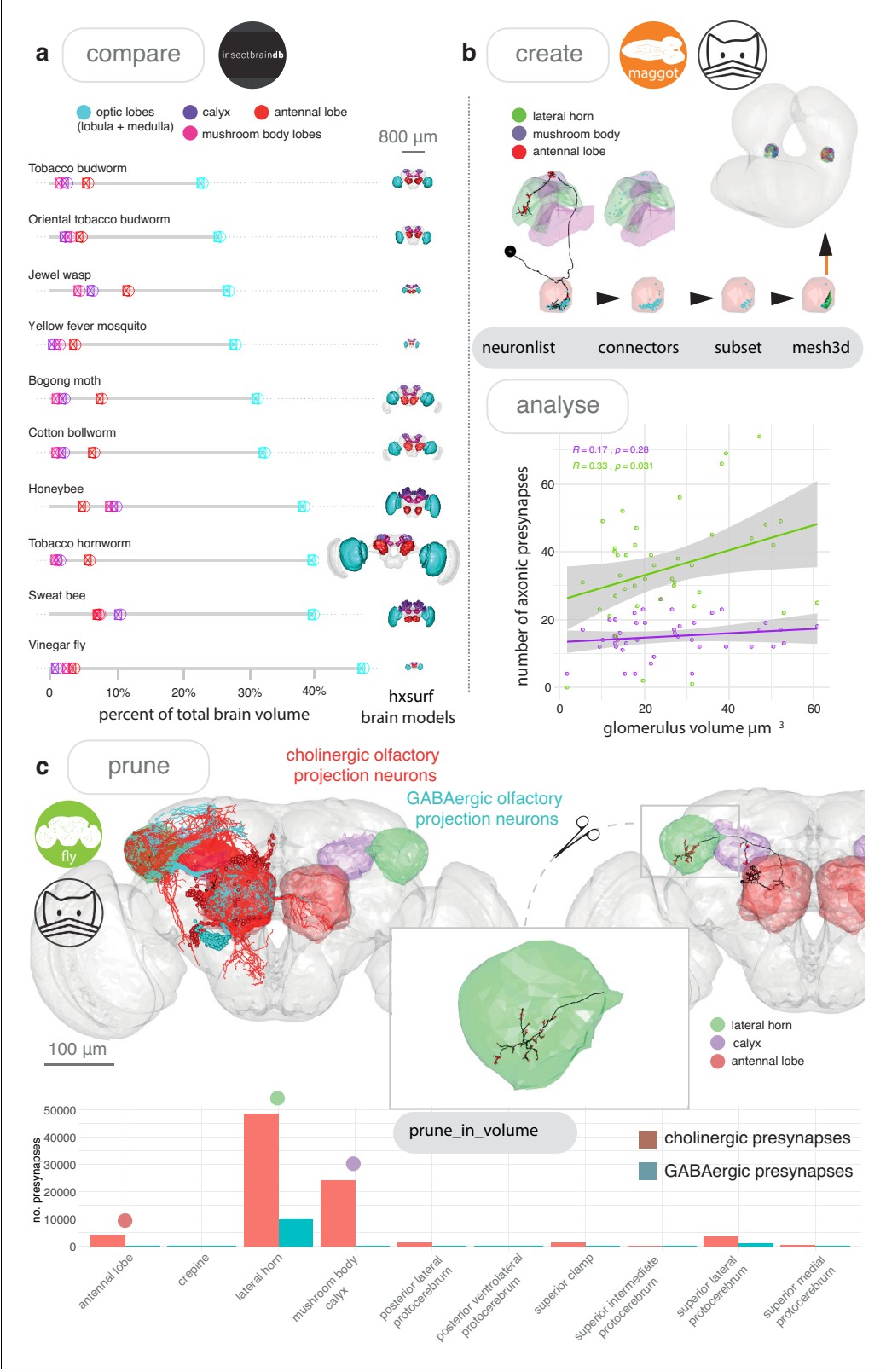

**Figure 3.** Neuroanatomical models with nat. (a) We accessed the InsectBrainDB.org via insectbrainr to obtain template brains for different species of insect (*Brandt et al., 2005*; *de Vries et al., 2017*; *El Jundi et al., 2018*; *Heinze and Reppert (2012)*; *Kurylas et al. (2008)*; *Løfaldli et al. (2010)*; *Stone et al. (2017)*; *Zhao et al., 2014*). The package insectbrainr converts retrieved OBJ files into hxsurf objects, which contain one set of 3D points for each whole brain, and then different sets of edges between these points to form 3D neuropil subvolumes. These subvolumes were already

*Figure 3 continued on next page*

*Figure 3 continued*

defined by expert annotators. Their volume are compared across insect brain, normalised by total brain size. Insect template brain data curated by: S. Heinze, M. Younger, J. Rybak, G. Pfuhl, B. Berg, B. el Jundi, J. Groothuis and U. Homberg. (**b**) We can create our own subvolumes by pulling synaptic neuron reconstructions (*Berck et al., 2016*) from a first-instar larva EM dataset (*Ohyama et al., 2015*) (a public CATMAID instance hosted by Virtual Fly Brain), extracting dendritic post synapses from olfactory projections neurons, and using synapse clouds from neurons of the same cell type, to define glomerular volumes by creating a bounding volume, i.e an α-shape or convex hull. Their volumes can then be calculated, and correlated with the number of presynapses the same neurons make in two higher-order brain regions, the lateral horn and the mushroom body calyx. (**c**) Volumes can be used to analyse skeleton data. In (**c**) we look again at olfactory projection neurons, this time from an adult fly EM dataset (*Zheng et al., 2018*) and use the `nat` function `pointsinside` with standard neuropil volumes (*Ito et al., 2014*) to find the numbers of presynapses GABAergic and cholinergic olfactory projection neurons from the antennal lobe make in different neuropils. These neuropils exist as a `hxsurf` object in our R package `nat.flybrains`.

The online version of this article includes the following figure supplement(s) for figure 3:

**Figure supplement 1.** Superxovel analysis with nat.

on their bilateral symmetry. *Figure 5d* shows 12 example neurons, taken from the bilateral subset of the FlyCircuit dataset, spanning the range of similarity scores from most asymmetric (A) to most bilaterally symmetric (L). Interestingly, the distribution of scores suggest that most bilateral neurons are reasonably symmetric.

It is also possible to use our mirroring registrations to test the degree of symmetry for sections of neurons. We take segments of a neuron and use our similarity metric to compute a score between the segment and the corresponding segment in the mirrored version of the neuron. This allows differences in innervation and axonal path between the two hemispheres to be clearly seen (*Figure 5e*).

## Bridging template spaces in *D. melanogaster*

Simply rescaling a sample image to match a reference brain usually fails due to differences in position and rotation (*Figure 6—figure supplement 1a*). An affine transformation can account for these differences, but not for differences in shape that may be of biological or experimental origin. To correct for these, we use a full non-rigid warping deformation, as described previously (*Jefferis et al., 2007*; *Rohlfing and Maurer, 2003*; *Rueckert et al., 1999*), see our Materials and methods. Briefly, a regular lattice of control points is created in the reference brain and corresponding control points in the sample brain are moved around to specify the deformations required to take the sample data into the reference space (*Figure 6c–g*). Deformations between control points are interpolated using B-splines, which define a smooth deformation of sample to reference (*Figure 6f*). The use of a mutual information metric based on image intensity avoids the requirement for landmarks to be added to each image – a time-consuming task that can often introduce significant inaccuracies. Our approach allows for the unsupervised registration of images and the independent nature of each registration allows the process to be parallelised across CPU cores. By utilizing a high-performance computational cluster, we re-registered, with high accuracy, the entire FlyCircuit dataset within a day.

Our bridging registrations can be deployed on any 3D `natverse`-compatible data using the function `xform_brain`. A successful and accurate bridging registration will result in the neuropil stains of two template spaces being well co-localised (*Figure 6*). After visually inspecting co-localised template spaces to check for any obvious defects, we find it helpful to map a standard neuropil segmentation (*Ito et al., 2014*) into the space of the new brain to check for more subtle defects (*Figure 6—figure supplement 2b*). If the registration passes these checks it can then be used to combine data from multiple datasets.

The creation of a bridge between a GAL4 expression library, such as the GMR collection (*Jenett et al., 2012*), and images of single neurons, such as those of FlyCircuit (*Chiang et al., 2011*), facilitates the decomposition of an expression pattern into its constituent neurons, allowing the correct assessment of innervation density on, for example, ipsilateral and contralateral sides (*Figure 6—figure supplement 2c*). Similarly, correspondences between neuroblast clones can be identified with co-visualisation. We bridge *Fru+* clones (*Cachero et al., 2010*) from IS2 space into the JFRC2 space of elav clones (*Ito et al., 2013*) and hence determine subset relations (*Figure 6—figure supplement 2b*). Furthermore, we can bridge the single neuron FlyCircuit data (*Chiang et al., 2011*) from the

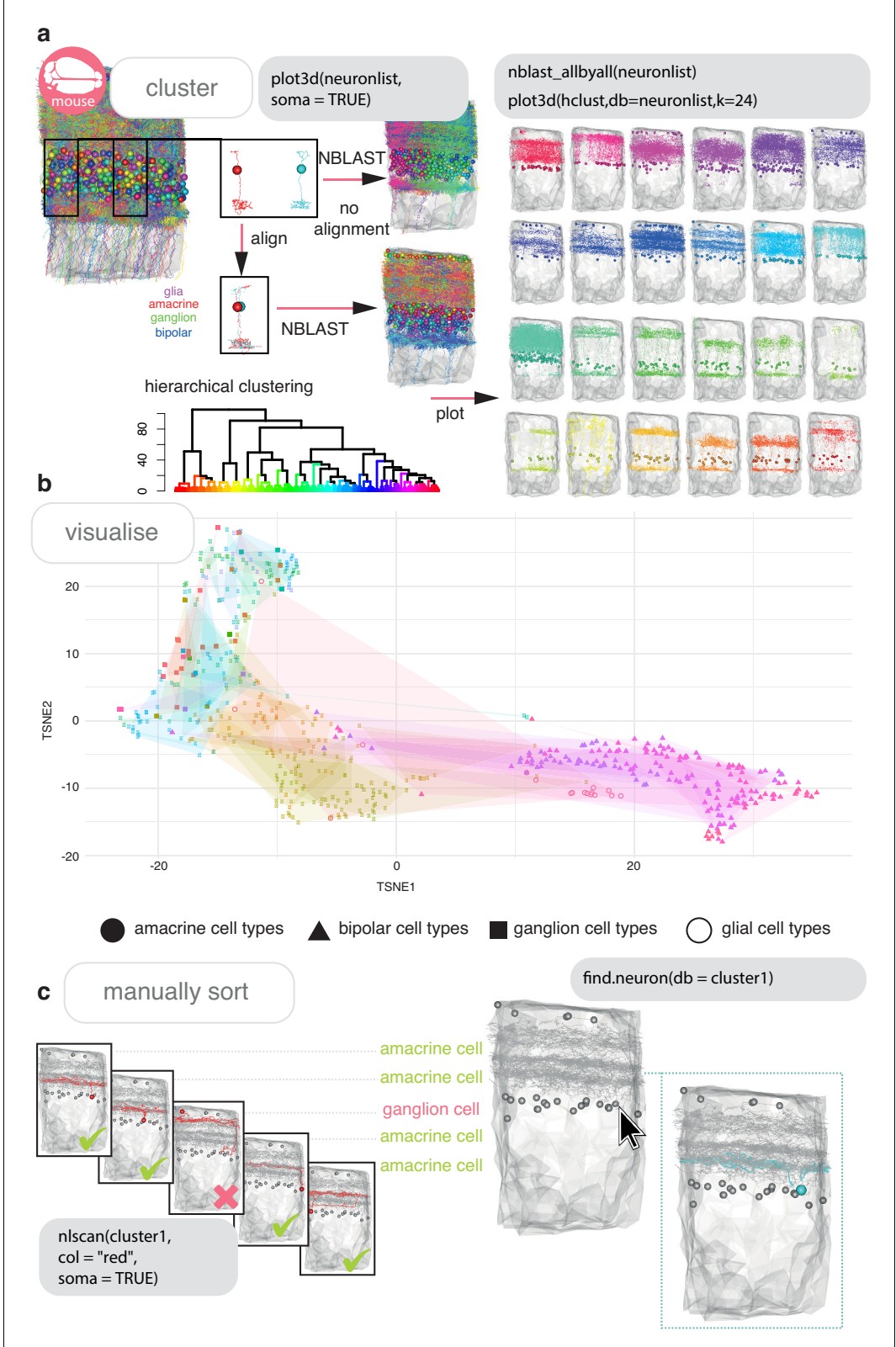

**Figure 4.** Cell typing with nat. (a) Neurons from a dense reconstruction from EM data of the mouse retina inner plexiform layer (*Helmstaedter et al., 2013*) can either be NBLAST-ed in situ (upper) or after alignment by their principal axes in 3D space (lower) in order to make a first pass at defining, finding or discovering morphological neuronal cell types using NBLAST. (b) A tSNE plot visualising the results of an aligned NBLAST of neurons in A,

*Figure 4 continued on next page*

*Figure 4 continued*

coloured by the manually annotated cells types seen in *Figure 2c*, with shapes indicating the cell class. (**c**) Manual sorting using the interactive `nat` functions `nlscan` and `find.soma` or `find.neuron` can help identify misclassifications and make assignments.

The online version of this article includes the following figure supplement(s) for figure 4:

**Figure supplement 1.** Cell typing zebrafish neurons with nat.
**Figure supplement 2.** Cell typing vinegar fly neurons with nat.

FCWB space into the IS2 space of the *Fru+* clones and use the known sexual dimorphisms of Fru clones to predict which neurons may be sexually dimorphic (*Figure 6—figure supplement 2c*).

The ability to bridge segmentations from one space to another is useful for checking innervation across datasets. While FlyCircuit single neurons (*Chiang et al., 2011*) were provided along with information on innervation density based on their own neuropil segmentation, this segmentation is not the same as the canonical one (*Ito et al., 2014*). We have bridged the latter segmentation into FCWB space and recalculated innervation for all the FlyCircuit neurons, providing a more standardised measure (*Figure 6—figure supplement 2g*). Further, we can compare neurons from FlyCircuit with those for which we have electrophysiological data (*Frechter et al., 2019*; *Kohl et al., 2013*), enabling us to suggest a functional role for unrecorded neurons based on their morphological similarity to recorded neurons (*Figure 6—figure supplement 2h*).

Both the FlyLight (*Jenett et al., 2012*) and Vienna Tiles libraries (*Tirian and Dickson, 2017*) contain a wealth of GAL4 lines amenable to intersectional strategies (*Luan et al., 2006*). However, as the two libraries are registered to different template spaces, it is difficult to predict which combinations of a FlyLight GMR line with a Vienna Tiles line would produce a good intersection (split-GAL4, targeting one cell type present in both parent lines) from the raw images provided by both. Bridging one library into the space of another (*Figure 6—figure supplement 2i*) enables direct co-visualisation (see also *Otsuna et al. (2018)* for an independent bridging output). This could be used manually or computationally to identify combinations that could potentially yield useful intersectional expression patterns (*Venken et al., 2011*).

It is also possible to warp 3D neuropils and neuron skeletons onto some target, without using landmark pairs. For this, Deformetrica (*Bône et al., 2018*; *Durrleman et al., 2014*) can be used to compute many pairwise registrations at once for different kinds of 3D objects to produce a single deformation of ambient 3D space describing a registration (*Figure 6—figure supplement 3*). This is a generic method that does not require landmark correspondences to be manually assigned. We give a simple example in *Figure 6—figure supplement 3a*, symmetrising a distorted brain and making a LM-EM bridge for first-instar larva, for which there is a nascent connectome (*Berck et al., 2016*; *Eichler et al., 2017*; *Ohyama et al., 2015*; *Schneider-Mizell et al., 2016*). With such a method it should be possible to bridge EM or LM data between developmental stages for a nervous system to make comparisons or identify neurons.

## EM to LM and back again

Finding neurons of the same cell type between a high-resolution EM dataset and light-level images of neurons (*Figure 7a*) is an essential step in identifying neurons and their genetic resources. So doing links connectivity and detailed morphology information acquired at the nanometer resolution to other forms of data. This can most easily be done by finding corresponding landmarks in EM data and a LM template space to build a registration (*Figure 6—figure supplement 1*).

In *Figure 7* and *Figure 8*, we give the general pipeline we used in recent publications (*Dolan et al., 2019*; *Dolan et al., 2018a*; *Frechter et al., 2019*; *Li et al., 2019*) to connect neurons sparsely labeled in a split-GAL4 line (registered to the template space JFRC2) to sparsely reconstructed neurons from an EM dataset (FAFB14). Neurons can be manually reconstructed (*Schneider-Mizell et al., 2016*) or, more recently, partially reconstructed by machine learning methods (*Januszewski et al., 2018*) as segments that can be manually concatenated (*Li et al., 2019*). A thin plate spline bridging registration between JFRC2 and FAFB14 was built by finding ~100 corresponding landmarks between the two brainspaces, for example the location of turns in significant tracts, the boundaries of neuropils, the location of easily identifiable single neurons (*Zheng et al., 2018*). This registration can be deployed using `xform_brain` and our `elmr` package.

**Table 1.** Neuron morphology resources currently available for the adult *D. melanogaster* brain.

| Dataset | Type | Count | Citations |
|---|---|---|---|
| FlyCircuit | Single neuron morphologies stochastically labeled from dense transmitter-related lines | ~23,000 neurons | (*Chiang et al., 2011*; *Shih et al., 2015*) |
| FlyLight GMR collection | Collection of genetic driver lines, driven by orthogonal transcription factors GAL4 (*Brand and Perrimon, 1993*) or (*Lai and Lee, 2006*) LexA | ~3500 GAL4 lines ~1500 LexA lines | (*Jenett et al., 2012*; *Pfeiffer et al., 2008*) |
| Vienna Tiles collection | Collection of genetic driver lines, driven by orthogonal transcription factors GAL4 or LexA | ~8000 GAL4 lines ~3000 LexA lines | (*Kvon et al., 2014*; *Tirian and Dickson, 2017*) |
| FlyLight split-GAL4 collection | Genetic driver lines labelling small constellations of neurons using the split-GAL4 system | ~400 sparse lines covering the mushroom body, lobula plate and columns, visual projection neurons, ellipsoid body, descending neurons, central complex, olfactory projection neurons (Y. Aso, personal communication, 2019) and lateral horn. | (*Aso et al., 2014*; *Aso and Rubin, 2016*; *Dolan et al., 2019*; *Klapoetke et al., 2017*; *Namiki et al., 2018*; *Robie et al., 2017*; *Wolff and Rubin, 2018*; *Wu et al., 2016*) |
| K. Ito, T. Lee and V. Hartenstein | Neuroblast clones for the central brain larval-born neurons, generated using the MARCM method (*Lee and Luo, 2001*) | ~100 neuroblast clones | (*Ito et al., 2013*; *Wong et al., 2013*; *Yu et al., 2013*) |
| FlyEM and Harvard Medical School | Volume-restricted connectomes | Hundreds of neurons from the mushroom body alpha lobe, two antennal lobe glomeruli and several columns of the optic medulla | (*Horne et al., 2018*; *Takemura et al., 2015*, *Takemura et al., 2013*, *Takemura et al., 2017*; *Tobin et al., 2017*) |
| FAFB project | Serial section transmission electron microscopy data for a single, whole adult female fly brain (*Zheng et al., 2018*), that has a partial automatic segmentation available (*Li et al., 2019*) | Raw image data for ~ 150,000 neurons of which several hundred have been partially reconstructed in recent publications, 7 thousand more unpublished; an estimated ~ 5% of neurons have some level of reconstruction. | (*Dolan et al., 2019*; *Dolan et al., 2018b*; *Felsenberg et al., 2018*; *Frechter et al., 2019*; *Huoviala et al., 2018*; *Sayin et al., 2019*; *Zheng et al., 2018*) |
| Various laboratories | Single neuron morphologies extracted from dye-filling (e.g. with biocytin) neurons | Hundreds across a range of studies, some cited here | (*Frechter et al., 2019*; *Grosjean et al., 2011*; *Jeanne et al., 2018*; *Jefferis et al., 2007*) |

**Table 2.** Exemplar *Drosophila* template brains.

| Template Brain | Description | Resources | DOI | Citation |
|---|---|---|---|---|
| Wuerzburg | Single nc82-stained female brain | - | - | (*Rein et al., 2002*) |
| TEFOR | Averaged brain generated from Rein et al. dataset (22, 22) | - | - | (*Arganda-Carreras et al., 2018*) |
| JRC2018F | A symmetrised high-quality template using brp-SNAP | - | 10.6084/m9.figshare.6825923 | (*Bogovic et al., 2018*) |
| Cell07 | Partial intersex nc82-stained averaged brain (14, 2) | ~240 lateral horn projection neuron tracings | 10.5281/zenodo.10570 | (*Jefferis et al., 2007*) |
| T1 | Intersex nc82-stained averaged brain | The Vienna Tiles collection | 10.5281/zenodo.10590 | (*Yu et al., 2010*) |
| IS2 | Intersex nc82-stained averaged brain | 1018 3D confocal images of fruitless neurons | 10.5281/zenodo.10595 | (*Cachero et al., 2010*) |
| FCWB | Intersex Dlg-stained averaged brain (17, 9) | Good for FlyCircuit data, ~16,000 neurons re-registered | 10.5281/zenodo.10568 | (*Costa et al., 2016*) |
| JFRC | Single nc82-stained female brain | The FlyLight collection | - | (*Jenett et al., 2012*) |
| JFRC2 | Spatially calibrated copy of JFRC | The FlyLight collection | 10.5281/zenodo.10567 | This study |
| IBN | Tri-labelled half brain, with n-syb-GFP | Neuropil and tract segmentations (half-brain) | - | (*Ito et al., 2014*) |
| IBNWB | Synthetic whole-brain version of IBN | Neuropil and tract segmentations (whole-brain) | 10.5281/zenodo.10569 | This study |
| FAFBV14 | An aligned volume for a single whole female fly brain from EM data | Thousands of single neuron partial manual reconstructions and fragmented automatic segmentation (*Li et al., 2019*) | - | (*Zheng et al., 2018*) |

By bridging multiple other light-level datasets into JFRC2 (*Figure 6*), candidate neurons from the EM brainspace can be co-visualised (*Figure 8c*) and NBLAST-ed against light-level datasets in order to confirm their cell type identity and consider results from different studies (*Chiang et al., 2011*; *Dolan et al., 2019*; *Frechter et al., 2019*; *Jeanne et al., 2018*; *Figure 8d*). However, FAFB14 contains unannotated image data for ~150,000 neurons (*Bates et al., 2019*), each requiring hours of manual reconstruction time, and person-power is limited. To find specific neurons in this volume, we can use the R package elmr to select a distinctive anatomical locus, for example the cell body fiber tract (*Frechter et al., 2019*) from 3D plotted neurons, and jump to its approximate coordinates in FAFB14 in a supported CATMAID instance using the generated URL (*Figure 7b*). Reconstruction efforts can then be focused at this location, being aware that the jump is not always completely accurate despite a good bridging registration as some light-level datasets can be ill-registered (*Figure 7b*). In the absence of an extant light-level reconstruction, candidate neurons can be found by identifying distinctive anatomical loci in the EM volume that correspond to the anatomy of the cell type in question (*Figure 7d*).

A user may also want to work the opposite way and connect an interesting EM reconstruction to light-level data, for example to identify a genetic resource that targets that neuron. In this situation, a similar pipeline can be used. For *D. melanogaster*, a reconstruction can be bridged into JFRC2 and NBLAST-ed against GAL4 lines (*Jenett et al., 2012*; *Tirian and Dickson, 2017*) read from image data and represented as vector clouds (*Costa et al., 2016*). Alternatively, image matching tools can be used, such as the recent colour depth MIP mask search (*Otsuna et al., 2018*), which operates as an ImageJ plug-in (*Figure 7c*).

Further, because close light-level matches for in-progress EM reconstructions reveal the likely morphology of non-reconstructed branches (*Figure 7c*) this process can help human annotators reconstruct neurons accurately and in a targeted manner, which may be desirable given how time intensive the task is. In order to further reduce this burden, we combined the natverse with a recent automatic segmentation of neurites in FAFB14 using a flood filling approach (*Li et al., 2019*),

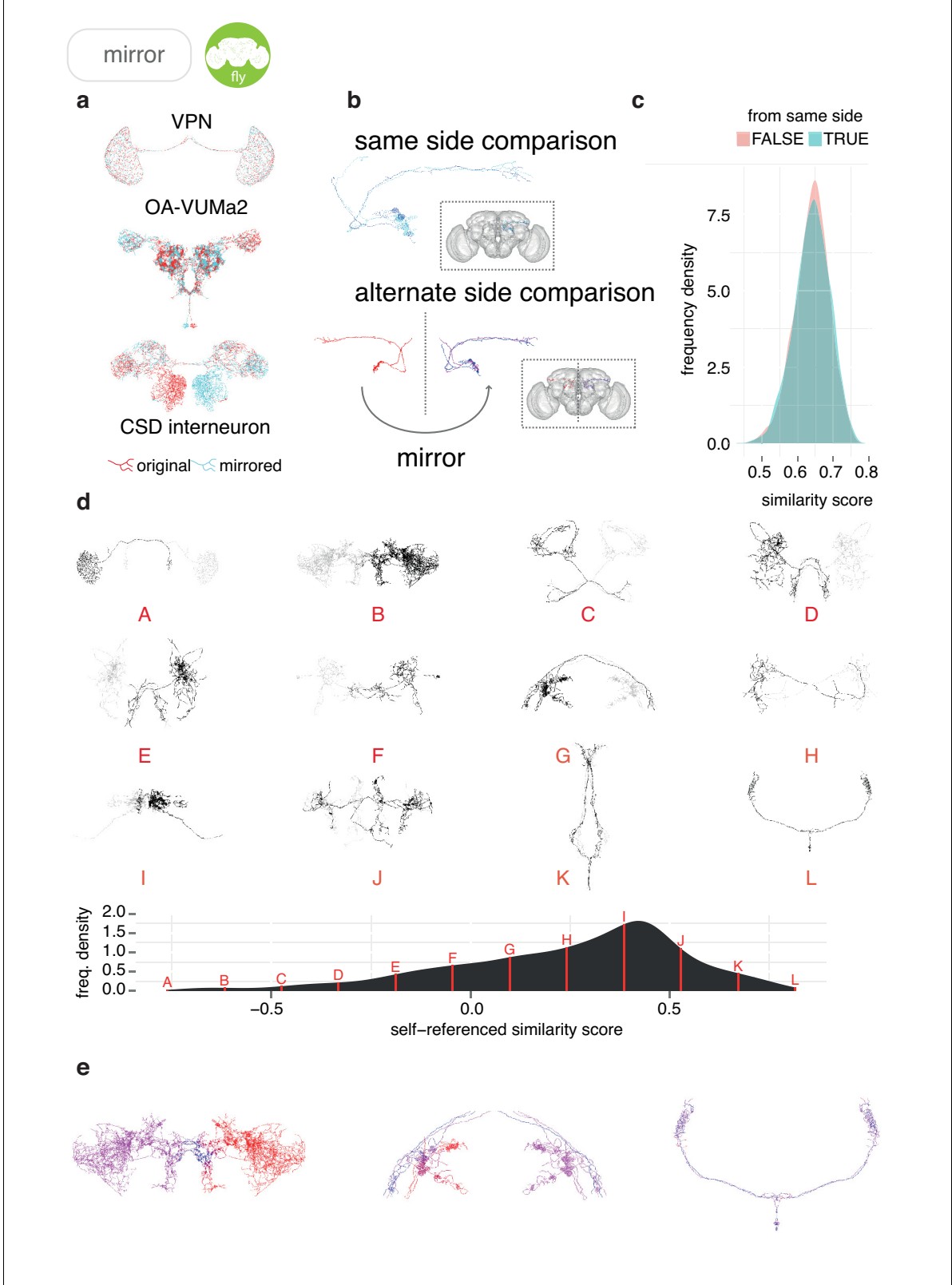

**Figure 5.** Sample applications of mirroring registrations. (**a**) Three FlyCircuit neurons along with mirrored versions; a visual projection neuron, OA-VUMa2 (**Busch et al., 2009**) and the CSD interneuron (**Dacks et al., 2006**). Co-visualisation facilitates the detection of differences in innervation, such as the higher density of innervation for the CSD interneuron in the lateral horn on the contralateral side compared to the ipsilateral lateral horn. (**b**) Neurons from the same side of the brain and alternate side of brain are compared and a similarity score generated. (**c**) Distributions of similarity scores

*Figure 5 continued on next page*

*Figure 5 continued*

for comparisons within the same brain hemisphere and across brain hemispheres. These scores are similar, because the mirroring registration is good. (d) Sequence of 12 example neurons (black) with mirrored counterparts (grey), having equally spaced similarity scores. Below, full distribution of scores for all neurons in FlyCircuit dataset. (e) Segment-by-segment measures of neuron similarity. Redder shades correspond to low degrees of symmetricity, bluer shades higher. Flipped version of neuron in gray.

The online version of this article includes the following figure supplement(s) for figure 5:

**Figure supplement 1.** Mirroring procedure.

which produces volumetric fragments of neurites, where segments may be fairly large, ~100 μm in cable length.

Our `fafbseg` package includes functions to implement improved up-/downstream sampling of neurons based on these segments, which we have recently discussed elsewhere (*Li et al., 2019*). We can also generate volumetric reconstructions of manually traced neurons by mapping them onto volumetric data (*Figure 8—figure supplement 1b*), hosted by a brainmaps server and visible through a Neuroglancer instance (*Figure 8—figure supplement 1a*). Currently, ~500 such segments will map onto one accurately manually traced neuron but only ~20 segments may constitute the highest Strahler order branches meaning that manual concatenation of these fragments speeds up discovery of coarse morphologies by ~10 x (*Li et al., 2019*). These fragments can be used to identify the neuron in question by NBLAST-ing against light-level data. Twigs and small-calibre, lower Strahler order branches are more difficult to automatically segment (*Figure 8—figure supplement 1d*). Nevertheless, matching tracings to segmentations allows us to estimate the volume of neurons that we have previously manually reconstructed (*Dolan et al., 2019*; *Dolan et al., 2018a*) by only tracing the neurites' midline (i.e. skeletonisation). We can therefore observe that superior brain neurons' axons are slightly thicker than their dendrites and their total cable length correlates strongly with neurite volumes (*Figure 8—figure supplement 1e*).

A densely reconstructed connectome, with ~35% of synapses connected up for just under half of the central fly brain has recently been made available by the FlyEM team at Janelia Research Campus (*Scheffer and Meinertzhagen, 2019*; *Shan Xu et al., 2020*). Neurons from this 'hemibrain' volume can be transformed to the JRC2018F light level template brain via a bridging registration constructed using the strategy described by *Bogovic et al. (2018)*. We have already wrapped this bridging registration within the natverse framework, thereby connecting it to the full network of fly template brains, datasets and analysis tools already described in this paper. We will release these tools when the hemibrain project makes its transforms publicly available.

## Discussion

The shape of a neuron is of major functional significance. Morphology is driven by and constrains connectivity. It is also the primary means by which neuroscientists have historically identified neuron classes. There have been three main drivers behind the recent emphasis on quantitative neuroanatomy: a) the ever increasing scale of new approaches for acquiring image data and reconstructing neurons, b) a drive to formalise descriptions of the spatial properties of neurons and networks at various scales, and c) a desire to intuit the organisational principles behind different nervous tissues and correlate these findings with dynamic data on neuron activity.

With the `natverse`, a suite of R packages for neuroanatomy with well-documented code and detailed installation instructions and tutorials available online, we aim to expedite analysis of these data in a flexible programming environment. The `natverse` allows a user to read data from local or remote sources into R, and leverage both `natverse` functions and the >10,000 R packages on CRAN (and more on Bioconductor, Neuroconductor, GitHub, etc.) to aid their data analysis. Users may also call `natverse` R functions from other languages such as Python, Julia, MATLAB. We have provided detailed examples to analyse skeleton and volume data from various sources and have made both R and Python code available at https://github.com/natverse/nat.examples. These examples demonstrate how to obtain skeleton and volume data, calculate basic metrics for neurons, examine synapses and other tagged biological features like microtubules, analyse morphology as a graph or through Strahler order and NBLAST searches, prune neurons, semi-manually cell type neurons, spatially transform neurons and create subvolumes using neurons. We have also given an

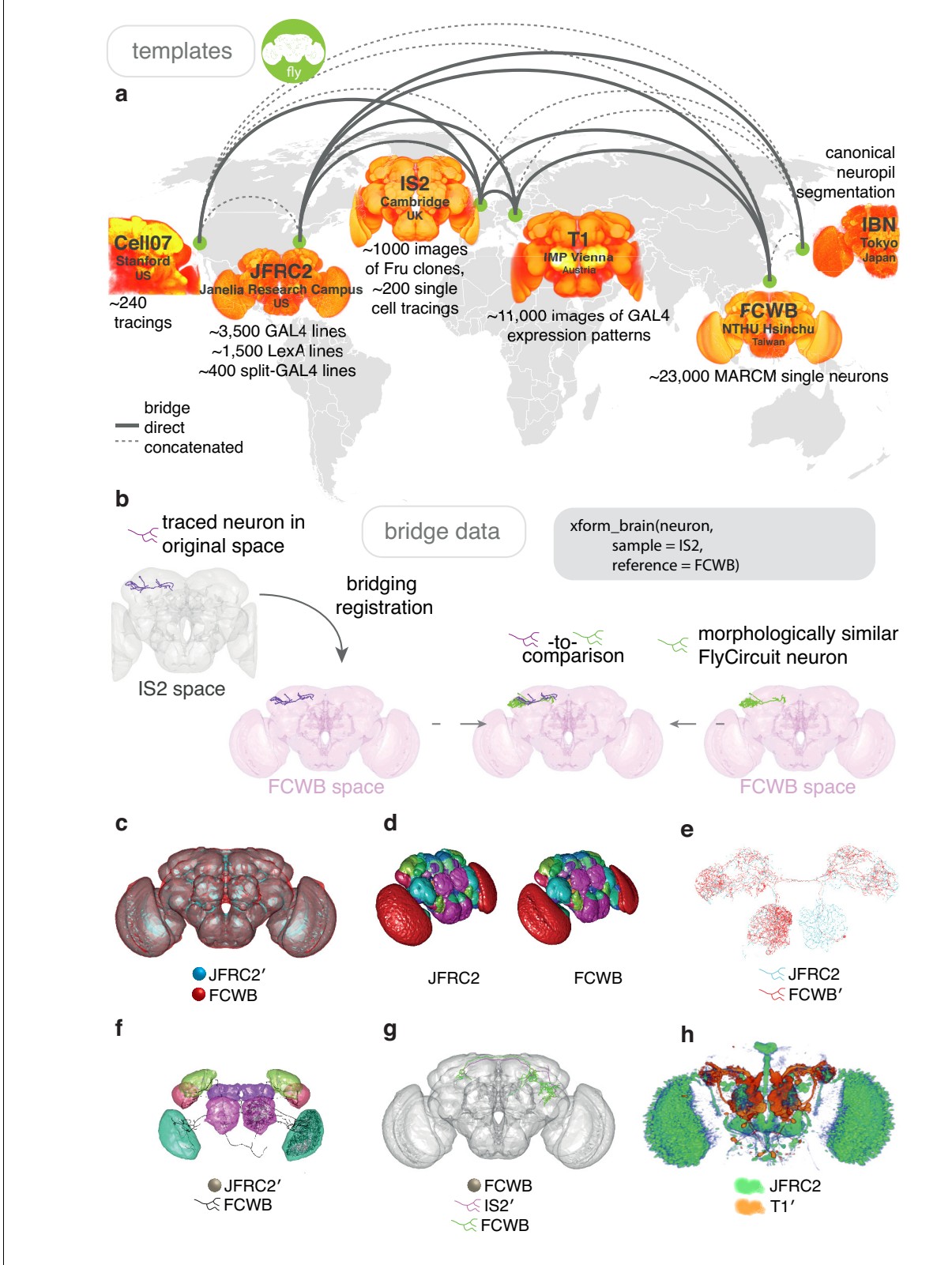

**Figure 6.** Bridging registrations for brain templates. (**a**) A small sample of *Drosophila* template brains used around the world are shown. (**b**) A partial neuron tracing (purple) made using Simple Neurite Tracer (*Longair et al., 2011*) being transformed (xform_brain) from the IS2 standard brainspace to the FCWB, where it can be co-visualised with a more complete, matching morphology from FlyCircuit, using our R package flycircuit. (**c**) Outermost neuropil boundaries for FlyCircuit (red) and FlyLight (cyan) template brains. Primed space names indicate original spaces for data. Unprimed

*Figure 6 continued on next page*

*Figure 6 continued*

space names indicate current space of image. (**d**) Neuropil segmentation from JFRC2 space alongside FCWB reformatted version. (**e**) CSD interneuron from FlyCircuit (red) and FlyLight, GMR GAL4 expression pattern (cyan). (**f**) Neuropil segmentation from JFRC2 (*Ito et al., 2014*) space that has been bridged into FCWB space, so it can be seen along with selected neurons from FlyCircuit. (**g**) A traced neuron in FCWB space alongside morphologically similar neuron from FlyCircuit. (**h**) Expression pattern of Vienna Tiles line superimposed on expression pattern of FlyLight line. (Since we made our bridging publically available in April 2014, *Otsuna et al., 2018* have also, separately, bridged these two datasets.).

The online version of this article includes the following figure supplement(s) for figure 6:

**Figure supplement 1.** Bridging procedure.
**Figure supplement 2.** Bridging examples.
**Figure supplement 3.** Warping registration without point-point correspondence.

example of building a more complex analysis, based on `natverse` tools but making use of other available R packages.

We hope that the `natverse` becomes a collaborative platform for which users can contribute to existing R packages or link to their own. We note that the `natverse` is an actively developing project and also anticipate a) an increasing interest in dealing with neurons as volumes as automatic segmentation of datasets becomes commonplace, b) expanding our bridging tools to support a wider range of species, and to map between similar species and developmental stages, c) writing libraries to facilitate the use of the `natverse` in other programming languages and toolboxes besides Python, and d) expanding the range of neurogeometric analysis algorithms readily available in the `natverse`.

In addition to general purpose `natverse` tools, we have generated some specific R packages to support ongoing projects in the *D. melanogaster* brain. We have constructed high-quality registrations for the bridging of data from one template space to another, along with registrations for mirroring data across brain hemispheres. In two of the largest cases, only raw unregistered data were available, so we began by registration to an appropriate template space. This has allowed us to deposit ~20,000 co-registered images from different sources in the virtualflybrain.org project. Averaged intersex template spaces can form high-quality registration templates for both sexes and we recommend the use of averaged brains to reduce the effects of sample-to-sample variation. We propose using a small number of template spaces, particularly those that are already associated with the most data (JFRC2) or of highest quality (*Bogovic et al., 2018*), as a hub. High-quality bridging registrations would be created between new template spaces and brains in the hub, ensuring that any template could be bridged to any other via appropriate concatenations and inversions of these registrations.

Using these resources, it is now possible to co-visualise and analyse more than 23,000 single neuron images (*Chiang et al., 2011*), expression patterns of >9500 GAL4 lines (*Jenett et al., 2012*; *Kvon et al., 2014*; *Tirian and Dickson, 2017*) and a near complete set of ~100 adult neuroblast clone lineage data (*Ito et al., 2013*; *Yu et al., 2013*) and easily combine these data with the standard insect brain name nomenclature system (*Ito et al., 2014*). For example we have calculated the neuropil overlap between single neurons in the FlyCircuit data, which we have deposited with virtualflybrain.org so they can be queried online. It will soon be possible to identify split-GAL4 lines, a synaptic EM reconstruction and the developmental clone of origin for any given neuron or neuronal cell type for *D. melanogaster*. We anticipate such mappings to become publicly available and easy to use via resources such as https://v2.virtualflybrain.org/. Significantly, if an experimenter is able to register their functional imaging data to a template brain space (*Mann et al., 2017*; *Pacheco et al., 2019*), or alternatively identify neuroanatomical features in that data that can be used to build a landmark-based affine or thin-plate spline registration (e.g. using `Morpho` *Schlager, 2017*), they may be able to directly link it to cell types discovered in other datasets, including EM datasets.

The near future will see generation of EM data for multiple whole adult Dipteran brains and larval zebrafish, possibly from different sexes and species, as well as quality automatic segmentations for such data's neurites (*Funke et al., 2019*; *Januszewski et al., 2018*) and synapses (*Heinrich et al., 2018*), even from anisotropic serial section transmission EM data (*Li et al., 2019*). Interpreting high-resolution EM connectomic data will be accelerated and enriched by making links to light level data (*Schlegel et al., 2017*). Furthermore, it is possible that connectomes and transcriptomes may be

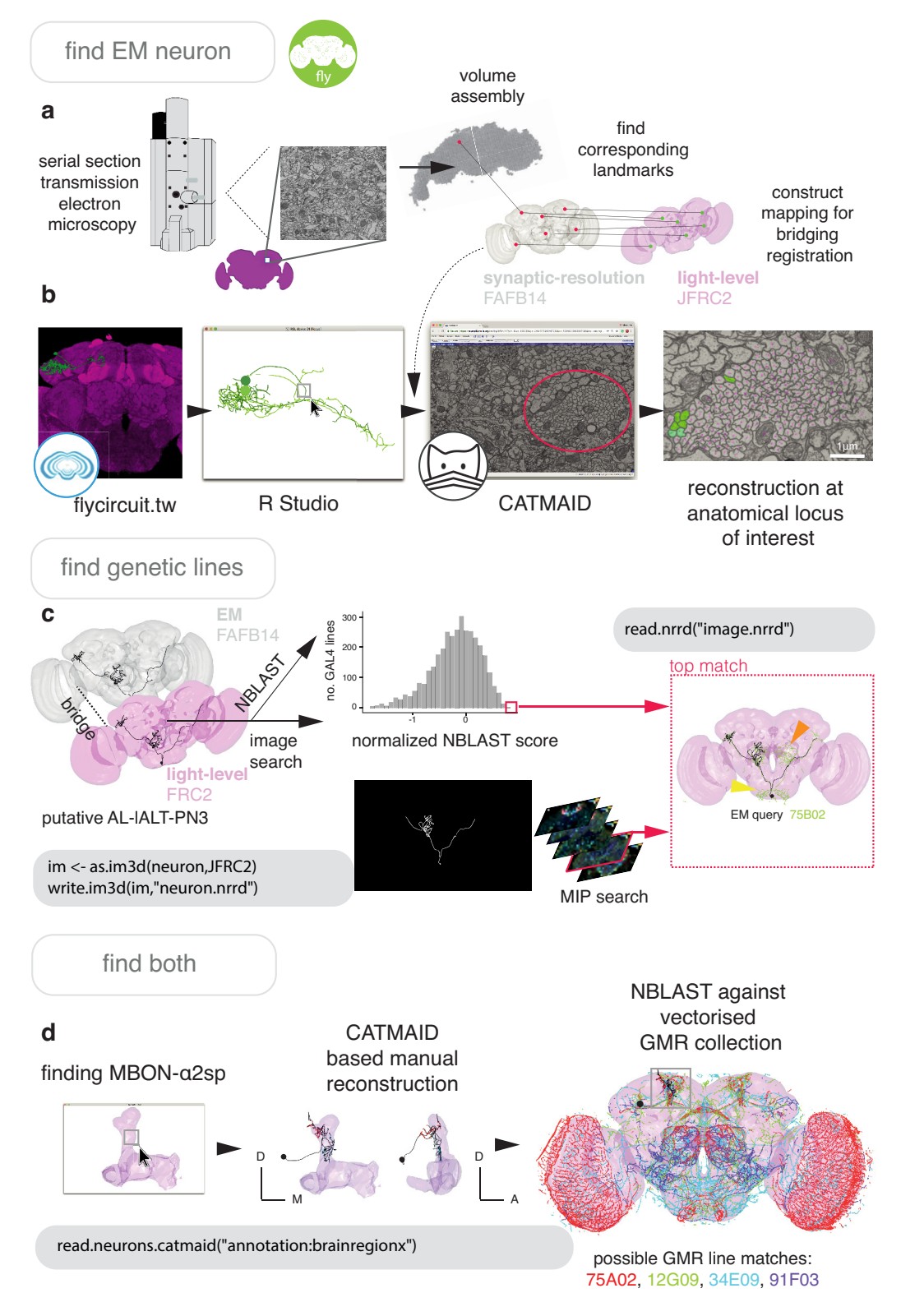

**Figure 7.** Finding specific neurons in EM and LM data. (a) Pipeline for acquiring EM neuron data. Serial section transmission EM at high speed with a TEM camera array (*Bock et al., 2011*) produced several micrographs per section at 4 × 4 nm resolution, ~40 nm thick. These were, per section, stitched into mosaics which were, across sections, registered to create the female adult fly brain v.14 template space (FAFB14, grey) (*Zheng et al., 2018*). Corresponding landmarks between FAFB14 and JFRC2 can be found and used to build a bridge. (b) The R package elmr can be used to select an

*Figure 7 continued on next page*

*Figure 7 continued*

anatomical locus, here the PD2 primary neurite tract (*Frechter et al., 2019*), from 3D plotted light-level neurons, taken from FlyCircuit, and generate a URL that specifies its correct coordinates in a FAFB14 CATMAID instance. Candidates (185) may then be coarsely traced out until they deviate from the expected light-level morphologies (178 pink dotted profiles, often a few minutes to an hour of manual reconstruction time to rule out neurons of dissimilar cell types sharing a given tract, similar cell types are more subtly different and might need to be near completely reconstructed). Those that remain largely consistent were fully reconstructed (green profiles, ~7–12 person-hours per neuron) (*Li et al., 2019*). (c) Close matches reveal likely morphology of non-reconstructed branches (orange arrow) but also contain off-target expression (yellow arrow). Identification of multiple candidate lines enables split-GAL4 line generation aimed at retaining common neurons in two GAL4 patterns. MultiColor FlpOut (MCFO) (*Nern et al., 2015*) of resultant splits can be compared with the EM morphology. Here, a candidate GAL4 line is found for AL-lALT-PN3 (*Frechter et al., 2019*; *Tanaka et al., 2012*) using NBLAST and a MIP search (*Otsuna et al., 2018*). (d) A recent dense, but volume-restricted reconstruction of the mushroom body α-lobe discovered a 'new' mushroom body output neuron type (MBON-α2sp) (*Takemura et al., 2017*). By bridging from the correct mushroom-body compartment using a mushroom body mesh (*Ito et al., 2014*) visualised in R Studio, to the FAFB14 EM data's equivalent space in CATMAID using the R package `elmr`, an experienced tracer can easily identify dendrites and find MBON-α2sp. By doing so, we found its previously unreported axon-morphology. We then imported the skeleton into R studio, bridged MBON-α2sp into the JFRC2 template space where it could be NBLAST-ed against GMR GAL4 lines to identify candidate lines containing the MBON.

---

linked on a cell type basis, using neuron morphology as a bridge (*Bates et al., 2019*). The `natverse` provides extensible functionality for easily combining and analysing all these data.

## Materials and methods

### R packages for neuroanatomy

The R programming language (*R Development Core Team, 2011*) is perhaps the premier environment for statistical data analysis, is well supported by the integrated development environment RStudio and is a strong choice for data visualisation (*Wickham, 2016*). It already hosts a wealth of packages for general morphometric and graph theoretic analysis (*Csardi and Nepusz, 2006*; *Duong, 2007*; *Lafarge et al., 2014*; *Schlager, 2017*). An R package is a bundle of functions, documentation, data, tests and example code (*Wickham, 2015*). R packages are discrete, standardised and highly shareable units of code. They are primarily installed either from the Comprehensive R Archive Network (CRAN, >14,000 packages, curated), Bioconductor (>1700 packages, curated) or GitHub (larger, uncurated), using just one or two function calls and an Internet connection. Confirmed stable versions of nat, nat.templatebrains, nat.nblast, nat.utils and nabor can be downloaded from the centralised R package repository, CRAN. The natmanager package provides a streamlined installation procedure and will advise the user if a GitHub account is required for the full natverse install (see http://natverse.org/install).

```
install.packages('natmanager')
# install core packages to try out the core natverse
natmanager::install('core')
# Full 'batteries included' installation with all packages
# You need a GitHub account and personal access token (PAT) for this
natmanager::install('natverse')
```

The R packages behind the `natverse` can be divided into four groups (*Figure 1A*):

#### Working with synaptic resolution data in nat

Group a) obtains synaptic-level data required for connectomes and includes `catmaid`, `neuprintr`, `drvid` and `fafbseg`. The package `catmaid` provides application programming interface (API) access to the CATMAID web image annotation tool (*Saalfeld et al., 2009*; *Schneider-Mizell et al., 2016*). CATMAID is a common choice for communities using terabyte-scale EM data to manually reconstruct neuron morphologies and annotate synaptic locations (*Berck et al., 2016*; *Dolan et al., 2018a*; *Eichler et al., 2017*; *Frechter et al., 2019*; *Ohyama et al., 2015*; *Zheng et al., 2018*). Users can use `catmaid` to read CATMAID neurons into R including the locations and associations of their synapses, and other tags that might identify biological entities such as somata, microtubules or gap junctions. Users can also leverage CATMAID's infrastructure of flexible hierarchical

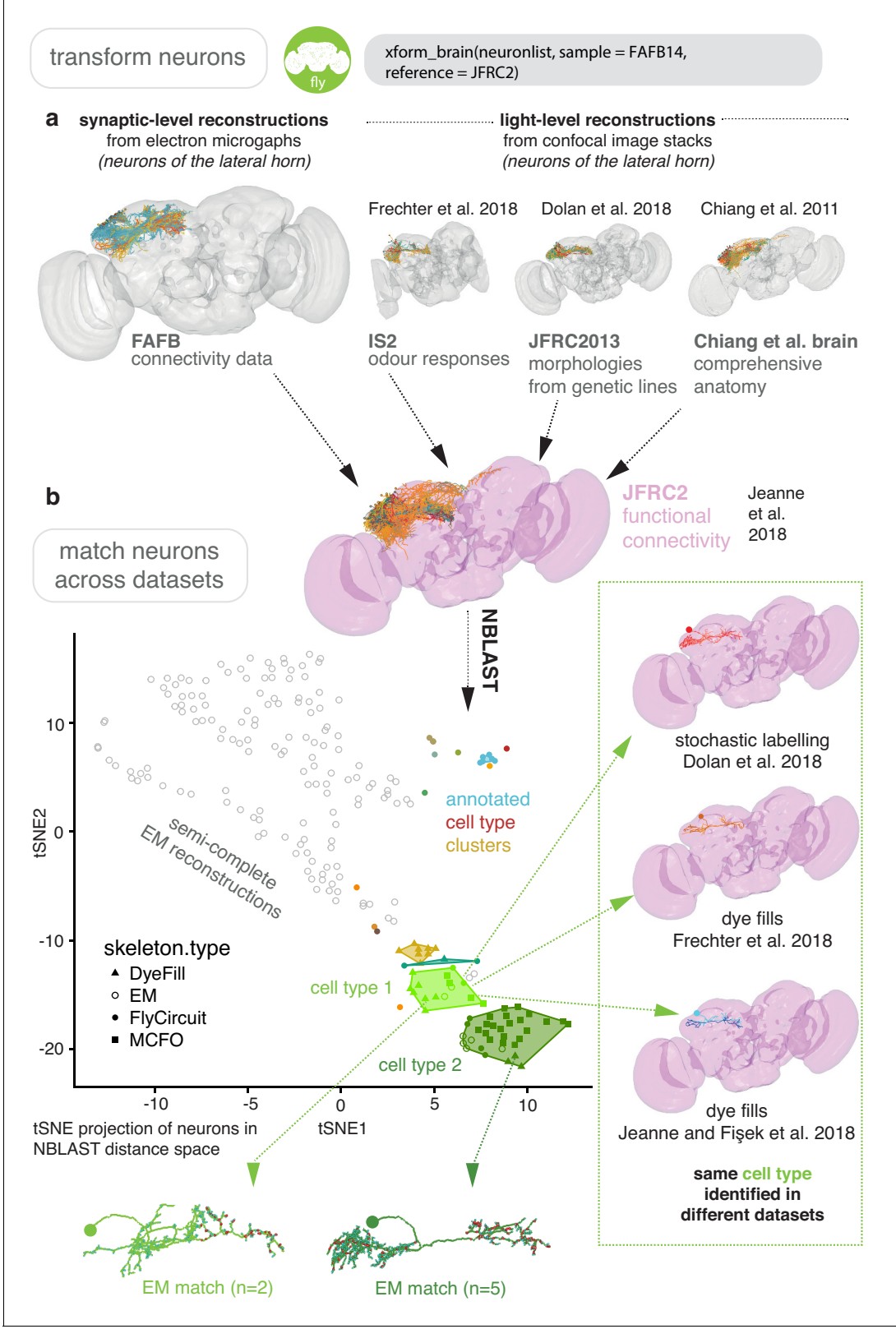

**Figure 8.** Bridging EM and LM data. (**a**) Sparse EM reconstruction providing a database of non-comprehensive, partial morphologies that can be searched using NBLAST. Candidate neurons from the EM brainspace can be NBLAST-ed against MCFO (*Nern et al., 2015*) data and other light-level datasets in order to connect them to cell-type-specific information, such as odour responses and functional connectivity (*Chiang et al., 2011*; *Dolan et al., 2019*; *Frechter et al., 2019*; *Jeanne et al., 2018*), by bridging these datasets into the same brainspace. (**b**) An all-by-all NBLAST of all

*Figure 8 continued on next page*

Figure 8 continued

neurons in the PD2 primary neurite cluster (*Frechter et al., 2019*) in multiple datasets can be shown as a tSNE plot. EM cell type matches can easily be found, as well as other correspondences between the light level datasets.

The online version of this article includes the following figure supplement(s) for figure 8:

**Figure supplement 1.** Using partial automatic segmentation of EM data.

semantic annotations to make queries for neurons for example in a brain region of interest. Further `catmaid` can edit CATMAID databases directly, for example by adding annotations, uploading neurons, synapses and meshes. Some CATMAID instances are kept private by a community before data publication. In this case, `catmaid` can enable a user to send authenticated requests to a CATMAID server, that is data can be kept private but still be read into R over an Internet connection. The packages `neuprintr` and `drvid` are very similar, except that they interact with API endpoints for different distributed annotation tools, the NeuPrint connectome analysis service (*Clements et al., 2020*; https://github.com/connectome-neuprint/neuPrint) and DVID (*Katz and Plaza, 2019*) and can retrieve neurons as volumes as well as skeletons. The package `fafbseg` aims to make use of the results of automatic segmentation attempts for large, dense brain volumes. It includes support for working with Google's BrainMaps and NeuroGlancer (https://github.com/google/neuroglancer). Automatic segmentation of EM data is a rapidly-developing field and this package is currently in active development; at present it only supports auto-segmentation (*Li et al., 2019*) of a single female adult fly brain (FAFB) dataset (*Zheng et al., 2018*).

## Working with light-resolution data projects in *nat*

Group b) is targeted at light microscopy and cellular resolution atlases, or mesoscale projectomes. Its packages, `neuromorphr`, `flycircuit`, `vfbr`, `mouselight`, `insectbrainr` and `fishatlas` can read from large repositories of neuron morphology data, many of which are co-registered in a standard brain space. `neuromorphr` provides an R client for the NeuroMorpho.org API (*Ascoli et al., 2007*; *Halavi et al., 2008*; *Nanda et al., 2015*), a curated inventory of reconstructed neurons (n = 107395, 60 different species) that is updated as new reconstructions are collected and published. Since its neurons derive from many different systems and species, there is no 'standard' orientation, and so they are oriented by placing the soma at the origin and aligning neurons by their principal components in Euclidean space. `insectbrainr` can retrieve neurons and brain region surface models from InsectBrainDB.org (n = 139 neurons, 14 species). Similarly `flycircuit` interacts with the flycircuit.tw project (*Chiang et al., 2011*; *Shih et al., 2015*), which contains >23,000 registered and skeletonised *D. melanogaster* neurons. The vfbr package can pull image data from VirtualFlyBrain.org, which hosts registered stacks of central nervous system image data for *D. melanogaster*, including image stacks for the major GAL4 genetic driver line collections (*Jenett et al., 2012*), neuroblast clones (*Ito et al., 2013*; *Yu et al., 2013*) and FlyCircuit's stochastically labelled neurons (*Chiang et al., 2011*). This non-skeleton data can be read into R as point clouds. The `fishatlas` package interacts with FishAtlas.neuro.mpg.de, which contains 1709 registered neurons from the larval *Danio rerio* (*Kunst et al., 2019*), while `mouselightr` does the same for the MouseLight project at Janelia Research Campus (*Economo et al., 2016*), which has generated >1000 morphologies. In both cases, investigators have acquired sub-micron single neuron reconstructions from datasets of whole brains using confocal (*Kunst et al., 2019*) or two-photon microscopy (*Economo et al., 2016*), modified tissue clearing techniques (*Treweek et al., 2015*), and generated a template brain with defined subvolumes.

## Working with registrations in nat

Group c) helps users make use of registration and bridging tools. The package `nat.ants` wraps the R package `ANTsRCore` (*Kandel et al., 2019*) with a small number of functions to enable `nat` functions to use Advanced Normalisation Tools (ANTs) registrations (*Avants et al., 2009*). The R package `deformetricar` does the same for the non-image (e.g. mesh or line data) based registration software Deformetrica (*Bône et al., 2018*; *Durrleman et al., 2014*) without the need for landmark correspondences. The `nat` package already contains functions to support CMTK registrations (*Rohlfing and Maurer, 2003*). The `nat.templatebrains` package extends `nat` to explicitly include

the notion of each neuron belonging to a certain template space, as well as functions to deploy bridging and mirroring registrations. Additionally, `nat.flybrains` contains mesh data describing commonly used template spaces for *D. melanogaster* as well as CMTK bridging and mirror deformations discussed in the latter half of the results section.

### Analysing data in nat

Group d) contains functions that help users to easily analyse neuron data as both skeletons and volumes. Its biggest contributor is `nat.` `nat.nblast` allows users to deploy the NBLAST neuron similarity algorithm (*Costa et al., 2016*), by pairwise comparison of vector clouds describing these neurons in R. Our `nabor` package is a wrapper for libnabo (*Elseberg et al., 2012*), a k-nearest neighbour library which is optimised for low dimensional (e.g. 3D) spaces. The package `elmr` is another fly focused package that has been born out of a specific use case. Currently, ~22 laboratories and ~100 active users worldwide are engaged with reconstructing *D. melanogaster* neurons from EM data (*Zheng et al., 2018*) using CATMAID (*Saalfeld et al., 2009*; *Schneider-Mizell et al., 2016*) in order to build a draft, sparse connectome. The package `elmr` allows users to read neurons from this environment, transform them into a template space where they can be compared with light-level neurons for which the community may have some other information (e.g. gene expression, functional characterisation, presence in genetic drive lines, etc.), then visualised and/or NBLAST-ed; all with only a few lines of code. This process enables CATMAID users to perform interim analyses as they reconstruct neurons, helping them to choose interesting targets for reconstruction and identify manually traced or automatically reconstructed neuron fragments (*Dolan et al., 2019*) or anatomical landmarks such as fiber tracts (*Frechter et al., 2019*), and so improve the efficiency of their targeted circuit reconstructions (*Dolan et al., 2018a*; *Felsenberg et al., 2018*; *Huoviala et al., 2018*).

## Building mirroring registrations

A simple 180° flip about the medio-lateral axis is insufficient to generate a left-right mirror for most neuroanatomical volumes; after flipping, the brain will not be perfectly centered in the image. It is first necessary to apply an affine registration to roughly match the flipped brain to the same location as the original. This results in a flipped brain with the correct gross structure (i.e. large structures such as neuropils align) but with mismatched fine details (e.g. bilaterally symmetric neurons may appear to innervate slightly different regions on either side (*Figure 5a*). For example, for the JFRC2 template space we found that points are, on average, displaced by 4.8 μm from their correct position, equivalent to 7–8 voxels of the original confocal image. The largest displacements, of the order of 10–15 μm, are found around the esophageal region (*Figure 5—figure supplement 1b*) and are likely due to specimen handling when the gut is removed during dissection. An ideal mirroring registration would result in zero total displacement after two applications of the mirroring procedure, that is a point would be mapped back to exactly the same location in the original brain hemisphere. Our constructed mirroring registrations have, on average, a round-trip displacement of less than a quarter of a micron — that is about the diffraction limit resolution of an optical microscope and less than half of the sample spacing of the original confocal image (*Figure 5—figure supplement 1c*).

## Building bridging registrations

Given a bridging registration A ↦ B, an attempt to produce the registration B ↦ A can be made via numerical inversion of the original registration. This is a computationally intensive process but we find it to be useful for neuroanatomical work as the inaccuracies are set by numerical error, which is much smaller than registration error. As the registration A ↦ B may be injective (i.e. points within brain A may map to a subset of the points within brain B), there may be some points in B, particularly near the boundaries of the brain, for which this inversion will not map them into A. To counter this we have, for some brains, constructed a new registration B ↦ A by explicitly registering B onto A, rather than relying on numerical inversion. Full details of the building of bridging registrations and their directions are shown in *Figure 6—figure supplement 1*. Here, the arrows indicate the direction of the forward transformation but, due to the ability to numerically invert the transformations, it is possible to travel 'backwards' along an arrow to transform in the opposite direction. While the inversion takes an appreciable time to calculate, the resulting errors are extremely small, far below the resolution of the original images, and only exist due to the finite precision with which the

floating-point numbers are manipulated. By inverting and concatenating bridging registrations as appropriate, it is possible to transform data registered to any of the template spaces to any of the other template spaces.

## Creating accurate registrations

Full, non-rigid warping registrations were computed using the Computational Morphometry Toolkit (CMTK), as described previously (*Jefferis et al., 2007*). An initial rigid affine registration with twelve degrees of freedom (translation, rotation and scaling of each axis) was followed by a non-rigid registration that allows different brain regions to move somewhat independently, subject to a smoothness penalty (*Rueckert et al., 1999*). In the non-rigid step, deformations between the independently moving control points are interpolated using B-splines, with image similarity being computed through the use of a normalised mutual information metric (*Studholme et al., 1999*). The task of finding an accurate registration is treated as an optimisation problem of the mutual information metric that, due to its complex nature, has many local optima in which the algorithm can become stuck. To help avoid this, a constraint is imposed to ensure the deformation field is spatially smooth across the brain, as is biological reasonable. Full details of the parameters passed to the CMTK tools are provided in the 'settings' file that accompanies each registration. To create mirroring registrations, images were first flipped horizontally in Fiji before being registered to the original template spaces using CMTK. For convenience, we also encoded the horizontal flip as a CMTK-compatible affine transformation, meaning that the entire process of mirroring a sample image can be carried in single step with CMTK.

## Construction of new template spaces

The template space provided by the FlyLight project (JFRC) is not spatially calibrated and so we added spatial calibration to a copy named JFRC2. Similarly, FlyCircuit images are registered to male and female template spaces and so we created an intersex template space from 17 female and 9 male brains to bring all FlyCircuit neurons into a common space, irrespective of sex. The IS2, Cell07 and T1 template spaces were left unaltered.

As the neuropil and tract masks provided by the Insect Brain Name working group (*Ito et al., 2014*) only cover half a brain (IBN), we extended the IBN template space into a new whole brain template (named IBNWB) to improve the quality of the bridging registration between the IBN files and the other whole brain templates. The green channel (n-syb-GFP) of the tricolour confocal data provided was taken, duplicated and flipped about the medio-lateral axis using Fiji (*Schindelin et al., 2012*). The Fiji plugin 'Pairwise stitching' (*Preibisch et al., 2009*) was used to stitch the two stacks together with an offset of 392 pixels. This offset was chosen by eye as the one from the range of offsets 385–400 pixels that produced the most anatomically correct result. The overlapping region's intensity was set using the 'linear blend' method. We attempted improving on this alignment using the Fourier phase correlation method that the plugin also implements, but this gave poor results – the algorithm favoured overlapping the optic lobes, with a half central brain being present on each of the left and right sides.

As the template space is synthesised from an affine transformation of the original IBN template, we only considered an affine bridging registration between IBN and IBNWB. The n-syb-GFP labelling used in the IBN template strongly labels a large collection of cell bodies close to the cortex, posterior of the superior lateral protocerebrum and lateral horn, that are not labelled by nc82 or Dlg and hence the warping registrations from IBNWB to the other whole brain templates are less accurate in this region.

## Construction of averaged template spaces

CMTK's `avg_adm` tool was used to iteratively produce new averaged seed brains given a set of template spaces and an initial seed brain drawn from the set. In each round, template spaces are registered to the seed brain and averaged to produce a new seed brain. After all rounds are complete, a final affine registration between the latest seed brain and a flipped version is calculated and then halved, resulting in a final brain that is centered in the middle of the image. The FCWB template was produced in this manner using 17 female and 9 male brains. We have developed documented tools to help users make average templates, here: https://github.com/jefferislab/MakeAverageBrain.

## Application of registrations to images, traced neurons and surface data

CMTK provides two commands, `reformatx` and `streamxform` that will use a registration to reformat images and transform points, respectively. The R package `nat` wraps these commands and can use them to transform neuroanatomical data, stored as objects in the R session, between template spaces. A 3D surface model of the standard neuropil segmentation (*Ito et al., 2014*) was generated from the labelled image stack, using Amira, read into R using nat, transformed into the different template template spaces, via JFRC2, and saved as new 3D surfaces. These can then be used to segment neurons in their original space, providing interesting volumetric data for a neuron such as the relative density of neuropil innervation.

## Flies

Wild-type (Canton S, Bloomington Stock Center, Indiana University) and transgenic strains were kept on standard yeast/agar medium at 25°C. Transgenics were a GH146-lexA line and the dFasciculin-II-GFP protein trap line (courtesy of M. Landgraf). Lines were balanced with CyO, Dfd-GMR-YFP or TM6b, Sb, Dfd-GMR-YFP balancer chromosomes (Bloomington Stock Center, Indiana University).

## Larval dissection, immunohistochemistry and imaging

Flies were mated a day before dissection and laid eggs on apple-juice based media with a spot of yeast paste overnight at 250C. Adults and large hatched larvae were subsequently removed, and small embryos (approx. the length of an egg) were dissected in Sorensen's saline (pH 7.2, 0.075 M). A hypodermic needle (30 ½ G; Microlance) was used to sever the mouth hooks of each larva, at which point the CNS extruded along with viscera, and was gently separated and stuck to a cover glass that has been coated with poly-L-lysine (Sigma-Aldrich) in a bubble of solution. The CNS' were then fixed in 4% formaldehyde (Fisher Scientific) in Sorensen's saline for 15 min at room temperature, and subsequently permeabilised in PBT (phosphate buffer with 0.3% Triton-X-100, SigmaAldrich). Incubated overnight in primary antibodies at 4°C and, after washes in PBT, in secondary antibodies for 2 hr at room temperature. Washes took place in either a bubble of fluid or shallow dish filled with solution to prevent collapse of brain lobes into the VNC. For this reason also, confocal stacks were acquired with a 40x dipping lens on a Zeiss LSM 710, voxel resolution $0.2 \times 0.2 \times 0.5$ microns. Primary antibodies used were Chicken anti-GFP (Invitrogen), 1: 10,000, mouse IgG1 anti-FasciclinII (DSHB), 1:10, rat N-Cadherin (DSHB) and mouse IgG1 Discs large-1, 1:50. Secondaries used were goat anti-mouse CF568, 1:600, goat anti-Chicken Alexa488, goat anti-mouse CF647, 1:600. Some antibodies and dissection training were kindly supplied by M. Landgraf.

## Visualisation

The majority of images shown in this manuscript were generated in R Studio. 3D images were plotted with `natverse` functions that depend on the R package `rgl` (*Murdoch, 2001*), 2D plots were generated using `ggplot2` (*Wickham, 2016*). 3D images of confocal data were visualised using Amira 6.0, and Paraview. Figures were generated using Adobe Illustrator.

## Data availability

The bridging and mirroring registrations are deposited in two version controlled repositories at http://github.com with revisions uniquely identified by the SHA-1 hash function. As some template spaces may have multiple versions, we identify each version by its SHA-1 hash as this is uniquely dependent on the data contained in each file. Since we use the distributed version control system, git, any user can clone a complete, versioned history of these repositories. We have also taken a repository snapshot at the time of the release of this paper on the publicly funded http://zenodo.org site, which associates the data with a permanent digital object identifiers (DOIs).To simplify data access for colleagues, we have provided spatially calibrated template spaces for the main template spaces in use by the *Drosophila* community in a single standard format, NRRD. These brain images have permanent DOIs listed in *Table 2*. We have also generated registrations for the entire FlyCircuit single neuron and FlyLight datasets. The registered images have been deposited at http://virtualfly-brain.org. The R packages `nat.flybrains` and `elmr` in the `natverse` also contain easy-to-use functions for deploying these registrations. The complete software toolchain for the construction and application of registrations consists exclusively of open source code released under the GNU

Public License and released on http://github.com and http://sourceforge.net. A full listing of these resources is available at http://jefferislab.org/si/bridging. All these steps will ensure that these resources will be available for many years to come (as has been recommended *Ito, 2010*).

## Acknowledgements

We are very grateful to the original data providers including Ann-Shyn Chiang, Gerry Rubin, Moritz Helmstaedter, Herwig Baier, Stanley Heinze, Arnim Jenett, Tzumin Lee, Kazunori Shinomiya and Kei Ito for generously sharing their image data with the research community. We specifically thank Arnim Jenett, Kazunori Shinomiya and Kei Ito for sharing the nc82-based *D. melanogaster* neuropil segmentation. We thank M-J Dolan for providing confocal microscopy exemplar images. Images from FlyCircuit were obtained from the NCHC (National Center for High-performance Computing) and NTHU (National Tsing Hua University), Hsinchu, Taiwan. We thank the Virtual Fly Brain team including MC, David Osumi-Sutherland, Robert Court, Cahir O'Kane and Douglas Armstrong for making some of our processed data available online through https://virtualflybrain.org. We note that data integration work with the virtualflybrain.org website was supported in part by an award from the Isaac Newton Trust to MC and Dr Cahir O'Kane. We thank Tom Kazimiers for help navigating the CATMAID API. We thank Alex Vourvoukelis, Alex von Klemperer, and Colin J Akerman for sharing unpublished reconstruction data.

We thank members of the Jefferis laboratory and the *Drosophila* Connectomics group for comments on this manuscript along with Jan Clemens, Jamie Jeanne and Stanley Heinze. We thank Jake Grimmett and Toby Darling for assistance with the LMB's computer cluster. This work made use of the Computational Morphometry Toolkit, supported by the National Institute of Biomedical Imaging and Bioengineering (NIBIB). We thank early users of the `natverse` for their help finding bugs and suggesting features, including but not limited to: István Taisz, Shanice Bailey, William Morris, Kathi Eichler, Dana Gallii, Sebastian Cachero, Erika Dona, Shahar Frechter, Konrad Heinz, Fiona Love, Paavo Huoviala, Amelia Edmondson-Stait and Lisa Marin.

This work was supported by the MRC (MC-U105188491), Starting and Consolidator grants (649111) from the European Research Council, and the Wellcome Trust (203261/Z/16/Z) to GSXEJ, the Boehringer Ingelheim Fonds and Herchel Smith Studentship (ASB) and a Fitzwilliam College Research Fellowship (JDM).

## Additional information

### Funding

| Funder | Grant reference number | Author |
| --- | --- | --- |
| Medical Research Council | MC-U105188491 | Alexander S Bates<br>James D Manton<br>Gregory SXE Jefferis |
| H2020 European Research Council | 649111 | Alexander S Bates<br>James D Manton<br>Marta Costa<br>Gregory SXE Jefferis |
| Wellcome | 203261/Z/16/Z | Sridhar R Jagannathan<br>Marta Costa<br>Philipp Schlegel<br>Gregory SXE Jefferis |
| Boehringer Ingelheim Fonds | | Alexander S Bates |
| Herchel Smith Fund | | Alexander S Bates |
| Fitzwilliam College, Univeristy of Cambridge | | James D Manton |

The funders had no role in study design, data collection and interpretation, or the decision to submit the work for publication.

## Author contributions
Alexander Shakeel Bates, James D Manton, Data curation, Software, Formal analysis, Validation, Investigation, Visualization, Methodology, Writing - original draft, Writing - review and editing; Sridhar R Jagannathan, Software, Investigation, Visualization, Methodology, Writing - review and editing; Marta Costa, Data curation, Software, Investigation, Methodology, Writing - review and editing; Philipp Schlegel, Software, Investigation, Methodology, Writing - review and editing; Torsten Rohlfing, Software; Gregory SXE Jefferis, Conceptualization, Data curation, Software, Formal analysis, Supervision, Funding acquisition, Validation, Investigation, Methodology, Project administration, Writing - review and editing

## Author ORCIDs
Alexander Shakeel Bates https://orcid.org/0000-0002-1195-0445
James D Manton https://orcid.org/0000-0001-9260-3156
Sridhar R Jagannathan https://orcid.org/0000-0002-2078-1145
Marta Costa http://orcid.org/0000-0001-5948-3092
Philipp Schlegel http://orcid.org/0000-0002-5633-1314
Gregory SXE Jefferis https://orcid.org/0000-0002-0587-9355

## Decision letter and Author response
Decision letter https://doi.org/10.7554/eLife.53350.sa1
Author response https://doi.org/10.7554/eLife.53350.sa2

# Additional files
## Supplementary files
• Transparent reporting form

## Data availability
All code is described at http://natverse.org/ which links to individual git repositories at https://github.com/natverse.

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
