## [Decision Letter]

Thank you for submitting your article "The natverse: a versatile toolbox for combining and analysing neuroanatomical data" for consideration by *eLife*. Your article has been reviewed by three peer reviewers, and the evaluation has been overseen K VijayRaghavan as the Senior and Reviewing Editor. The reviewers have opted to remain anonymous.

The reviewers have discussed the reviews with one another and the Reviewing Editor has drafted this decision to help you prepare a revised submission.

Summary:

This manuscript documents a large number of computational neuroanatomy software packages, and associated workflow pipelines. Many of these tools are potentially useful for neuroanatomy work in a variety of species. That said, the need for these tools has been largely driven by the availability of large public neuroanatomy datasets from the *Drosophila* brain at single-cell (and often single-synapse) spatial resolution; therefore, it is appropriate that the authors focus their descriptions here on the workflow of the *Drosophila* neurobiologist, with additional frequent specific instances drawn from other species. Many of the tools described here have already been released publicly by the authors, and they have proven to be useful to many other researchers, as evidenced by the citations they have already accrued. Now the authors have augmented these existing tools with new ones that extend the functionality of the existing tools. This manuscript describes and rationalizes the entire workflow that these tools enable. Broadly speaking, this workflow allows researchers to assemble brain-wide network diagrams with relative ease – beginning with a cell (or cell-type) of interest, researchers can use these tools to identify the upstream and downstream connections of that cell, to locate similar cell types, to identify and classify morphological variants of those cells, to search for genetic driver lines to target cells of interest and their upstream and downstream partners, and to analyze the subcellular distribution of presynapses and postsynapses within cells of interest. Among the key elements here are the bridging registrations that bring different datasets into same 3D coordinate space, allowing researchers to leverage the enormous (and rapidly growing) amount of detailed single-cell and single-synapse-scale neuroanatomical data in *Drosophila*. This type of computational neuroanatomy workflow is currently revolutionizing *Drosophila* neurobiology, due in large part to the work of these authors. Without it, each individual lab would write their own custom methods to accomplish a fraction of what is possible in the natverse – the authors should be commended for developing these tools and making them accessible to the community quickly. Overall, this suite of methods represents an incredibly important tool for many disciplines within neuroscience. They are already having a transformative impact on the field of *Drosophila* systems/circuits neuroscience, and the deployment of these tools for mouse and fish datasets is highly valuable.

Once the manuscript is appropriately revised we would strongly support publication.

Essential revisions:

1) As this is a tools and resources paper, we would like to make a suggestion that might help – to borrow from their own language – the casual user. In the Results section, the authors lay out the various kinds of analyses possible with the natverse suite of tools. Reading the paper, what can be done is increasingly impressive. It might be nice to have this laid out in the front end of the paper in some form. Take, for example, subsection “Bridging template spaces in *D. melanogaster*” that deals with bridging template spaces – performing in silico intersections of Gal4 lines will help a researcher narrow their search for clean lines. Or in subsection “EM to LM and back again” – researchers are often trying to find 'their neuron' described by light microscopy in an EM volume and vice versa. It will be very nice to state these and other possibilities in the beginning giving the casual user a flavour of what can be done before diving into the details.

2) Skeletonization of neurons: the authors have been leading the way in development of these neuroanatomy tools and have had to make choices about how to connect between different data sets. The choice of skeletonizing a neuron was a useful one, but it is clear that one loses valuable information by skeletonization. Connectome projects are generating beautiful flood filled reconstructions of neurons (e.g. see Li et al., 2019) – the authors develop fafbseg to connect flood-filled segments together based on a manually-traced skeleton. But, it would be useful for the authors to discuss how their EM to LM to EM tools could be adapted to starting with neuron volumes (that they get from fafbseg or from other connectome projects) versus skeletons. They mention this issue very briefly in the Discussion, but don't address how well these tools would work on volume data, or whether new tools would need to be developed, and what the constraints would be.

3) Many users of these resources are generating in vivo neural imaging data and it will be increasingly important to register in vivo brain activity and templates to fixed brain templates (for example, see Mann et al., 2017 or Pacheco et al., 2019). It would be useful for the authors to discuss this issue and to provide such registrations.

4) It would be useful for the authors to generate a small number of example pipeline codes, with some explanation in the code for each step. A small repository with data to use with these example pieces of code will allow a first-time user to run the code locally and get results in a short time, and then to use these pieces of code as a starting point for further analyses. The examples shown in the paper cover all of the basic analysis examples, and adding code that one can simply run to generate the figures shown in the paper, will help new users (if this is included somewhere, apologies)

5) In some cases, a short video with example use will make it much easier to understand and use the tools appropriately. For example for the functions 'nlscan' and 'plot3D'. Generally, short example videos are useful. Such videos already exist for NBLAST and VFB.

6) Having one-line installation is definitely useful and saves time, but a few steps are still needed before running it – this is OK, but please describe them to save troubleshooting time. This reviewer tried installation on two machines, and got the same error on both – API rate limit exceeded for xx. Perhaps a description like this would help:

Install R, Rstudio

Create a Github account if you don't have one install.packages("usethis") usethis::browse_github_pat() usethis::edit_r_environ() then update the environment variable with the personal access token, following the instructions in Rstudio - copy your personal access token as instructed

Paste to.…\.Renviron

Should have, for example, these two lines:

GITHUB_PAT=xx

---

## [Author Response]

Essential revisions:1) As this is a tools and resources paper, we would like to make a suggestion that might help – to borrow from their own language – the casual user. In the Results section, the authors lay out the various kinds of analyses possible with the natverse suite of tools. Reading the paper, what can be done is increasingly impressive. It might be nice to have this laid out in the front end of the paper in some form. Take, for example, subsection “Bridging template spaces in *D. melanogaster*” that deals with bridging template spaces – performing in silico intersections of Gal4 lines will help a researcher narrow their search for clean lines. Or in subsection “EM to LM and back again” – researchers are often trying to find 'their neuron' described by light microscopy in an EM volume and vice versa. It will be very nice to state these and other possibilities in the beginning giving the casual user a flavour of what can be done before diving into the details.

This is a good suggestion. We have added a new paragraph at the end of the Introduction so that the reader has an overview of the key applications discussed in the Results.

2) Skeletonization of neurons: the authors have been leading the way in development of these neuroanatomy tools and have had to make choices about how to connect between different data sets. The choice of skeletonizing a neuron was a useful one, but it is clear that one loses valuable information by skeletonization. Connectome projects are generating beautiful flood filled reconstructions of neurons (e.g. see Li et al., 2019) – the authors develop fafbseg to connect flood-filled segments together based on a manually-traced skeleton. But, it would be useful for the authors to discuss how their EM to LM to EM tools could be adapted to starting with neuron volumes (that they get from fafbseg or from other connectome projects) versus skeletons. They mention this issue very briefly in the Discussion, but don't address how well these tools would work on volume data, or whether new tools would need to be developed, and what the constraints would be.

We have amended the subsection “Neuron skeleton data” to address neuron volume data briefly,  and state in the Discussion that updating the natverse to work with, and implement specific algorithms for, volume representations for neurons is a future goal (Discussion, second paragraph). We agree with the reviewer that it will become increasingly important as volume data for EM neurons becomes more prevalent.

3) Many users of these resources are generating in vivo neural imaging data and it will be increasingly important to register in vivo brain activity and templates to fixed brain templates (for example, see Mann et al., 2017 or Pacheco et al., 2019). It would be useful for the authors to discuss this issue and to provide such registrations.

We feel that it is out of the scope of the current work (which already covers a wide range of applications) to provide an example of registering functional imaging data. However, the reviewer has raised an important point, and we have now adapted the penultimate paragraph of the Discussion to note how one might link functional imaging data to, for example, EM data.

4) It would be useful for the authors to generate a small number of example pipeline codes, with some explanation in the code for each step. A small repository with data to use with these example pieces of code will allow a first-time user to run the code locally and get results in a short time, and then to use these pieces of code as a starting point for further analyses. The examples shown in the paper cover all of the basic analysis examples, and adding code that one can simply run to generate the figures shown in the paper, will help new users (if this is included somewhere, apologies).

This is included in our GitHub repo: https://github.com/natverse/nat.examples. We have reworded part of the fourth paragraph of the Introduction, to make the nature of this code a little clearer.

5) In some cases, a short video with example use will make it much easier to understand and use the tools appropriately. For example for the functions 'nlscan' and 'plot3D'. Generally, short example videos are useful. Such videos already exist for NBLAST and VFB.

The reviewer makes a good point. We have now included Videos 1-5 (20 minutes in total) that demonstrate some of our functions in RStudio, giving examples for plotting neurons, bridging neurons between template brains, using NBLAST and comparing EM and LM data. We will also add this to the natverse website. The video can also be found as a single 20 min file here:

https://drive.google.com/open?id=1wBpAG4V9bBujdLcryWk8P-K7YTpgfZTO

6) Having one-line installation is definitely useful and saves time, but a few steps are still needed before running it – this is OK, but please describe them to save troubleshooting time. This reviewer tried installation on two machines, and got the same error on both – API rate limit exceeded for xx. Perhaps a description like this would help:Install R, RstudioCreate a Github account if you don't have one install.packages("usethis") usethis::browse_github_pat() usethis::edit_r_environ() then update the environment variable with the personal access token, following the instructions in Rstudio - copy your personal access token as instructedPaste to.…\.RenvironShould have, for example, these two lines:GITHUB_PAT=xx

This is a very good point that we had largely overlooked during testing, since even our users without a natverse installation had already authenticated with GitHub. We have now provided what we hope is a significantly improved installation procedure which should remove the need for GitHub authentication in most situations, while helping the user through the setup process listed above if necessary. We also provide more detailed installation instructions covering this point. Finally we also suggest a simplified install of “core” packages that bypasses this requirement altogether. We have updated our Materials and methods (subsection “R packages for Neuroanatomy”) as well as the instructions at http://natverse.org/install. Essentially

install.packages("natmanager")

# Then basic install.… natmanager::install("core")

# OR full install

natmanager::install("natverse")

should handle most install cases without requiring the end user to authenticate with GitHub.